# Skill-Pro: Learning Reusable Skills from Experience via Non-Parametric PPO for LLM Agents

**Qirui Mi** [1]  **Zhijian Ma** [2 3]  **Mengyue Yang** [4]  **Haoxuan Li** [5]  **Yisen Wang** [5]  **Haifeng Zhang** [2]  **Jun Wang** [6]

## Abstract

LLM-driven agents excel at sequential decision-making but often rely on on-the-fly reasoning, re-deriving solutions even in recurring scenarios. This insufficient experience reuse leads to computational redundancy and instability. To bridge this gap, we propose **Skill-Pro**, a framework enabling agents to autonomously learn reusable procedural skills from interaction experiences without parameter updates. By formalizing a **Skill-MDP**, Skill-Pro transforms passive episodic narratives into executable Skills defined by activation, execution, and termination conditions to ensure executability. To achieve reliable reusability without capability degradation, we introduce **Non-Parametric PPO**, which leverages semantic gradients for high-quality candidate generation and a PPO Gate for robust Skill verification. Through score-based maintenance, Skill-Pro sustains compact, high-quality procedural memory. Experimental results across in-domain, cross-task, and cross-agent scenarios demonstrate that Skill-Pro achieves superior reuse rates and significant gains with extreme memory compression. Visualized evolutionary trajectories and Skill distributions further reveal how Skill-Pro transparently accumulates, refines, and reuses procedural knowledge to facilitate long-term autonomy.

## 1. Introduction

Large Language Model (LLM)-driven agents have shown strong performance in complex sequential decision-

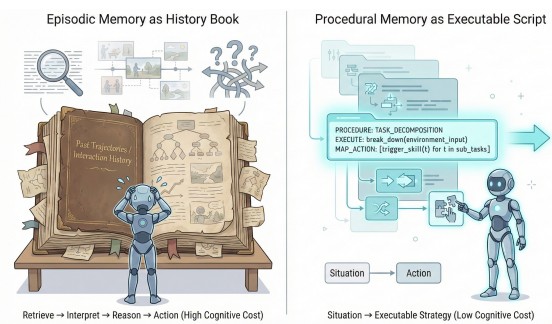

*Figure 1.* **Episodic memory versus procedural memory in LLM-driven agents.** Episodic memory retrieves past interactions for reference, requiring inference-heavy reasoning at decision time. Procedural memory encodes reusable procedural Skills that directly map situations to actions, enabling efficient experience reuse.

making (Park et al., 2023; Shinn et al., 2023). However, this performance is often driven by **on-the-fly reasoning**, where agents interpret prompts, observations, and feedback in real-time to generate solutions (Wei et al., 2022; Yao et al., 2022; 2023). Even in recurring situations, they typically redo the full reasoning process from scratch, treating each interaction as an unseen problem. This insufficient experience reuse results in substantial computational redundancy and increases the risk of error accumulation in long-horizon scenarios, eventually leading to lower reliability in execution (Liu et al., 2024; Press et al., 2023).

To incorporate interaction experience, existing work broadly falls into two paradigms. **Parametric methods**, such as Reinforcement Learning (RL), attempt to encode experience into model parameters (Sutton et al., 1998; Ouyang et al., 2022). While effective in specific domains, these approaches face high training costs, risks of catastrophic forgetting, and a narrowing of general-purpose capabilities (Kirkpatrick et al., 2017). Alternatively, **non-parametric methods** improve behavior at inference time without updating the base LLM, most commonly via external memory (Hu et al., 2025). These agents store diverse forms of experience in external memory, including past trajectories (Lewis et al., 2020; Rajesh et al., 2025), distilled reflections (Xu et al., 2025; Zhao et al., 2024), and structured graphs (Zhang et al., 2025a; Xia et al., 2025) or workflows (Wang et al., 2024). At decision time, they retrieve stored experiences to condition the LLM's reasoning and improve performance, typically

---

[1]Key Laboratory of Interdisciplinary Research of Computation and Economics, Shanghai University of Finance and Economics [2]Institute of Automation, Chinese Academy of Sciences [3]School of Artificial Intelligence, Chinese Academy of Sciences [4]University of Bristol [5]Peking University [6]University College London. Correspondence to: Jun Wang <jun.wang@ucl.ac.uk>, Haifeng Zhang <haifeng.zhang@ia.ac.cn>.

without updating the base LLM. However, despite their effectiveness, these methods predominantly operate as forms of episodic memory (Cohen & Squire, 1980), storing and retrieving past experiences as "history books" to be consulted (Fig. 1). Even with large memory, the agent still has to spend its limited context window interpreting retrieved cases and re-deriving solutions, effectively returning to the inference-heavy loop. Inspired by *procedural memory* in human cognition, an implicit system that directly maps situations to action patterns (Squire, 2004); once acquired, it enables the automatic execution of Skills without conscious re-derivation (Anderson, 1982). While frameworks like Claude Agent Skills (Anthropic, 2025) reuse manually encoded procedures, this work investigates *how LLM agents can autonomously learn reusable procedural Skills from interaction experience for future decision-making.*

However, establishing reusable procedural memory faces three fundamental obstacles: **C1: Executability.** Interaction experience is often stored as passive episodic narratives describing past events rather than active decision procedures that can be directly instantiated at runtime. **C2: Reusability.** The challenge lies in ensuring that stored procedures can be reliably invoked and reused in future tasks while providing a consistent performance gain. **C3: Non-Parametric Optimization.** The difficulty lies in Learning Reusable Skills through non-parametric methods while preserving the agent's general-purpose capabilities.

To address these challenges, we propose **Skill-Pro**, a framework designed to learn reusable **pro**cedural **Skill**s from interaction experience without parameter updates. First, Skill-Pro formalizes procedural units as Skills consisting of Activation Conditions, Execution Procedures, and Termination Conditions. By constructing a **Skill-MDP**, the agent selects and reuses these executable procedures (Skills) for decision-making to ensure executability (**C1**). To achieve reliable reusability (**C2**) through non-parametric optimization (**C3**), we introduce **Non-Parametric PPO**. This mechanism leverages *semantic gradients* extracted from batch trajectories to propose refined Skill candidates, while a PPO-style Trust-Region Verification (PPO Gate) ensures the selection of high-quality Skills for inclusion in the procedural memory. Furthermore, an online scoring mechanism filters out redundant or low-quality procedures. Experimental results demonstrate that Skill-Pro achieves **superior reuse rates, significant performance gains, and extreme memory compression** compared to baselines across in-domain, cross-task, and cross-agent scenarios (Table 1, 2). Ablation studies confirm that Semantic Gradients and the PPO Gate are indispensable for generating and verifying high-quality skills, while online scoring preserves long-term evolutionary gains (Table 3, Fig. 3). Finally, visualized evolutionary trajectories (Fig. 4) and Skill distribution (Fig. 5) reveal how Skill-Pro's procedural memory is transparently constructed and reused to facilitate long-term autonomy. **Our core contributions are three-fold:**

- **Procedural Skill Formalization**: We introduce the Skill-MDP, transitioning LLM agents from episodic narratives to reusable procedural Skills.
- **Non-Parametric PPO**: We propose a parameter-free optimization mechanism leveraging *Semantic Gradients*, a *PPO Gate*, and *score-based maintenance* to evolve high-quality skills without weight updates.
- **Superior reuse rates and performance gain** across diverse scenarios with extreme memory compression.[1]

## 2. Related Work

*(A comprehensive literature review is shown in Appendix A.)*
**Learning from Interaction.** LLM agents improve decision-making via *parametric fine-tuning*, such as reinforcement learning (Ouyang et al., 2022; Rafailov et al., 2023; Guo et al., 2025), or *non-parametric adaptation*. While parametric updates yield strong performance, they often incur high computational costs and risk catastrophic forgetting or overspecialization (Ziegler et al., 2020; Shi et al., 2025a; Luo et al., 2025), driving the shift toward memory-augmented agents as a more efficient, non-parametric alternative.

**Memory-Augmented LLM Agents.** Existing frameworks primarily differ in experience representation: (i) **Episodic Trajectories:** Storing raw trajectories for case-based reasoning (Park et al., 2023); (ii) **Abstracted Knowledge:** Distilling experience into summaries (Yang et al., 2025), high-level principles (Wu et al., 2025; Agrawal et al., 2025; Cai et al., 2025), or failure-derived insights (Zhao et al., 2024); (iii) **Structured & Compressed Memory:** Organizing experience into graphs (Zhang et al., 2025a; Jimenez Gutierrez et al., 2024; Rezazadeh et al., 2024; Xia et al., 2025), dense vectors (Das et al., 2024; Zhang et al., 2025b), or dynamic knowledge snippets (Asai et al., 2024; Shi et al., 2025b; Zhou et al., 2025b); (iv) **Workflow:** Maintaining explicit task-completion paths (Wang et al., 2024). Skill-centric and procedural frameworks (Wang et al., 2023; Tan et al., 2024; Zhu et al., 2023; Sumers et al., 2023; Han et al., 2025; Fang et al., 2025) leverage executable logic, yet robust reusability remains non-trivial. To bridge the gap between storage and reusability, we propose **Skill-Pro** to learn procedural memory for efficient, autonomous long-term execution.

## 3. Reusable Procedural Units: Skills

In this section, we introduce *Skills* as reusable procedural units integrated into the decision-making process of LLM agents. We define a **Skill** as a temporally extended procedural units specifying: (1) *when* it should be activated, (2) *how* the agent should execute a sequence of actions, and (3)

---

[1]Code is available at:
https://github.com/Miracle1207/Skill-Pro

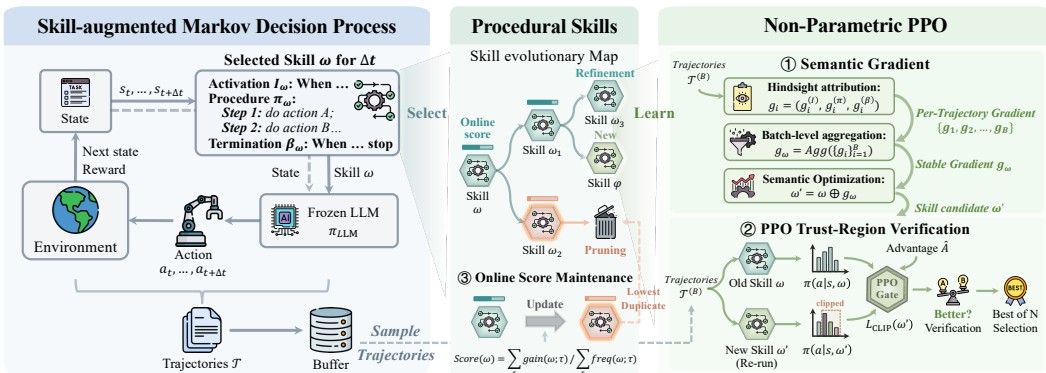

Figure 2. **Overview of the Skill-Pro framework. (Left) Skill-MDP:** The agent selects a Skill $\omega$ based on state $s_t$ and activation conditions. A frozen LLM executes $\omega$ into primitive actions $a_t$ over multiple steps until termination. Post-episode trajectories $\mathcal{T}$ are stored in a buffer. **(Middle) Procedural Skills:** Skills are dynamically managed via *refinement*, *generation*, and *score-based pruning* to maintain pool quality. **(Right) Non-Parametric PPO:** Evolution proceeds in two stages: ① **Semantic Gradient**: Derives and aggregates per-trajectory gradients through hindsight attribution to generate candidates $\omega'$. ② **PPO Gate**: Filters candidates via trust-region verification, admitting only the best-performing valid candidate into the Skill pool.

*when* control should return.

Unlike human procedural memory (Cohen & Squire, 1980; Squire, 2004), which is largely implicit, our Skills are currently represented in an explicit, readable form, similar to systems such as Claude Agent Skills (Anthropic, 2025). While explicit in representation, Skills can be hidden at the system level during execution; directions toward implicit procedural representations are discussed in Appendix F.1.

### 3.1. Problem Formulation

We formulate the LLM agent's decision-making process as a **Skill-augmented Markov Decision Process (Skill-MDP)**. The Skill-MDP extends the classical Markov Decision Process (MDP) by introducing a *dynamic Skill pool* $\Omega$, which explicitly represents the agent's procedural memory and organizes decision making around the selection and execution of reusable procedural Skills. Formally, a Skill-MDP is defined as a tuple

$$\mathcal{M}_\Omega = (\mathcal{S}, \mathcal{A}, \Omega, P, R, \gamma),$$

where $\mathcal{S}$ denotes the semantic state space and $\mathcal{A}$ denotes the primitive action space, both represented in natural language; $\Omega = \{\omega^{(1)}, \ldots, \omega^{(K)}\}$ denotes the pool of available Skills, which serves as the agent's procedural memory, with $K$ Skills stored; $P$ and $R$ are the state transition and reward functions, respectively, and $\gamma \in (0, 1]$ is the discount factor.

In a Skill-MDP, at each time step $t$, the agent first selects a Skill via a Skill-selection policy $\mu$, conditioned on the current state $s_t$ and the current Skill pool $\Omega$:

$$\omega_t \sim \mu(\omega \mid s_t, \Omega).$$

Conditioned on the active Skill $\omega_t$, the agent then generates a primitive action using an LLM-driven action policy:

$$a_t \sim \pi_{\text{LLM}}(a \mid s_t, \omega_t). \quad (1)$$

Accordingly, the hierarchical policy over Skills and primitive actions factorizes as

$$\pi_\Omega(\omega_t, a_t \mid s_t) = \mu(\omega_t \mid s_t, \Omega)\,\pi_{\text{LLM}}(a_t \mid s_t, \omega_t), \quad (2)$$

After executing $a_t$, the agent receives reward $r_t$ and transitions to next state $s_{t+1}$ until the horizon $T$ is reached. While Eq. (2) shares a similar factorization with memory-augmented agents (e.g., Memento (Zhou et al., 2025a)), which typically optimize the retrieval policy $\mu$, our work focuses on autonomously evolving the Skill pool $\Omega$.

**Optimization Objective.** We aim to optimize the agent's decision-making performance under the hierarchical policy $\pi_\Omega$, measured by the expected cumulative discounted return:

$$\max_{\pi_\Omega} \mathbb{E}\left[\sum_{t=0}^{T} \gamma^t r_t\right]. \quad (3)$$

In this paper, we focus on learning the Skill pool $\Omega$, while keeping the LLM policy and the Skill-selection mechanism fixed. We next specify the internal structure of a Skill, the basic reusable procedural unit in the Skill-MDP.

### 3.2. Skill Structure

We define a *Skill* $\omega \in \Omega$ as a reusable, natural-language procedural unit that specifies *when* to activate, *how* to act while active, and *when* to return control. This enables Skills to be reused across similar states. Formally, a Skill is defined as the tuple

$$\omega = \langle \mathcal{I}_\omega, \pi_\omega, \beta_\omega \rangle,$$

representing its activation condition, execution procedure, and termination condition.

**(1) Activation Condition $\mathcal{I}_\omega$.** The activation condition specifies when a Skill is invoked. Rather than a learned classifier in latent space, $\mathcal{I}_\omega$ is a natural-language description of observable context patterns where the Skill applies. During

decision making, the Skill-selection policy $\mu$ uses $\mathcal{I}_\omega$ to select Skills based on the current state or interaction history.

**(2) Execution Procedure $\pi_\omega$.** The execution procedure $\pi_\omega$ specifies an ordered sequence of actions to be executed while the Skill $\omega$ is active, expressed as in natural language. At each time step $t$, the LLM generates a primitive action conditioned on the current state $s_t$ and $\pi_\omega$ (Eq. (1)), enabling the agent to reuse procedural steps without re-deriving deliberative reasoning from scratch.

**(3) Termination Condition $\beta_\omega$.** The termination condition specifies when the execution of a Skill should end. Like the activation condition, $\beta_\omega$ is expressed in natural language and evaluated on the current state:

$$\beta_\omega(s_t) = 1 \ \text{ iff } s_t \text{ satisfies } \beta_\omega.$$

If $\beta_\omega(s_t) = 1$, the Skill $\omega$ terminates and $\mu$ selects the next Skill; otherwise, it remains active.

**Example.** We illustrate with an example Skill. A detailed case showing how this Skill guides primitive action generation during execution is provided in Appendix D.2.

---

**Name:** StrategicPlanning

**Activation Condition $\mathcal{I}_\omega$:** *When the task begins and no prior information or feedback is available.*

**Execution Procedure $\pi_\omega$:**

*Step 1: Establish an initial hypothesis space based on the task constraints.*

*Step 2: Generate a diverse exploratory action that maximally reduces uncertainty.*

**Termination Condition $\beta_\omega$:** *Terminate after the first exploratory action is executed and feedback is observed.*

---

### 3.3. Skill Selection

At each decision step $t$, the agent maintains a Skill pool $\Omega_t$. A new Skill is selected by the Skill-selection policy $\mu$ only upon termination of the current Skill. In that case, $\mu$ selects a Skill $\omega_t \in \Omega_t$ based on the current state $s_t$ and the pool $\Omega_t$. We present two simple instantiations of $\mu$.

**(i) Selection by Similarity.** This mechanism selects the Skill whose activation condition best matches the state $s_t$:

$$\omega_t = \arg\max_{\omega \in \Omega_t} \text{Sim}(s_t, \mathcal{I}_\omega),$$

where $\text{Sim}(s_t, \mathcal{I}_\omega)$ measures the similarity between the current state and the activation condition $\mathcal{I}_\omega$. The similarity function $\text{Sim}(\cdot, \cdot)$ can be implemented via cosine similarity over embeddings, or LLM as judge.

**(ii) Selection by Value.** We also support value-based selection to prioritize Skills with higher expected return. We first form a candidate set of the top-$k$ Skills by similarity:

$$\Omega_t^{(k)} = \text{TopK}_{\omega \in \Omega_t} \text{Sim}(s_t, \mathcal{I}_\omega),$$

where $\Omega_t^{(k)} \subseteq \Omega_t$ and $|\Omega_t^{(k)}| = k$. We then select the Skill with the highest estimated value in this set:

$$\omega_t = \arg\max_{\omega \in \Omega_t^{(k)}} Q(s_t, \omega).$$

Here $Q(s_t, \omega)$ denotes an estimate of the expected return obtained by invoking Skill $\omega$ from state $s_t$.

These mechanisms are simple instantiations of $\mu$ and can be replaced by more advanced Skill retrieval policies, e.g., RL-based methods (Zhou et al., 2025a; Zhang et al., 2026). Our focus lies on Skill reuse and evolution.

### 3.4. Skill Pool Evolution

The Skill pool is learned from interaction experience and evolves over time. In our framework, the pool is updated using trajectories generated under the current policy. Given a batch of trajectories $\mathcal{T}^{(B)}$ collected under the Skill-augmented policy $\pi_\Omega$, we define a Skill pool evolution operator $\mathcal{E}$:

$$\Omega_{\text{new}} = \mathcal{E}(\Omega_{\text{old}}, \mathcal{T}^{(B)}), \tag{4}$$

which synthesizes new Skills, refines existing Skills, and prunes those that underperform empirically.

This paper focuses on Skill pool evolution. The LLM action policy $\pi_{\text{LLM}}$ and the Skill-selection policy $\mu$ are kept fixed, and learning proceeds through repeated application of $\mathcal{E}$. Under this setting, optimizing Eq. (3) reduces to optimizing the evolution operator $\mathcal{E}$:

$$\max_{\mathcal{E}} \ \mathbb{E}_{\tau \sim \pi_{\Omega^*}} \left[ \sum_{t=0}^{T} \gamma^t r_t \right], \quad \text{where } \Omega^* = \mathcal{E}^{(N)}(\Omega_0). \tag{5}$$

Here, $\mathcal{E}^{(N)}$ denotes $N$ successive applications of $\mathcal{E}$, each using newly collected experience.

## 4. Non-Parametric PPO for Skill Evolution

In this section, we present a Proximal Policy Optimization (PPO)-inspired non-parametric method for Skill pool evolution, which we refer to as *Non-Parametric PPO*. This method leverages PPO-style trust-region principles to realize reliable Skill pool evolution as defined in Eq. (4), without updating any LLM parameters.

Standard PPO (Schulman et al., 2017) improves a parameterized stochastic policy through gradient-based optimization of a clipped surrogate objective. In contrast, our Non-Parametric PPO replaces parameter updates with Skill refinement, and consists of two components: (1) generating candidate Skills via *semantic gradients* extracted from trajectories, and (2) accepting candidates only if they satisfy a *PPO-style trust-region verification* under the frozen LLM policy. The complete procedure is shown in Algorithm 1.

## 4.1. Semantic Gradients

To learn without updating LLM parameters, we introduce *Semantic Gradients* as learning signals that specify how a Skill should be refined.

**Per-trajectory semantic gradients.** Unlike TextGrad (Yuksekgonul et al., 2024), which uses automatic differentiation to optimize static variables for instantaneous response quality, our *semantic gradients* are designed for sequential decision making. These gradients provide natural-language update directions extracted from interaction trajectories, indicating how a Skill's activation, execution, and termination conditions should be refined via *hindsight attribution*. Consider a Skill $\omega = \langle \mathcal{I}_\omega, \pi_\omega, \beta_\omega \rangle$ and trajectories $\mathcal{T}^{(B)}$ where it is invoked. For each trajectory $\tau_i$, we analyze the segment controlled by $\omega$ and attribute the outcome to its activation condition, execution procedure, or termination condition. This yields a structured semantic gradient

$$g_i = \nabla_{\text{sem}}(\tau_i, \omega) = \left( g_i^{(\mathcal{I})},\ g_i^{(\pi)},\ g_i^{(\beta)} \right).$$

where each component is a natural-language refinement suggestion for the corresponding Skill component. Intuitively, $g_i$ serves as a local update direction for Skill $\omega$ induced by trajectory $\tau_i$. An example of semantic gradients from our experiments is given in the Appendix D.1.

**Batch-level aggregation.** Individual trajectories may provide inconsistent update signals. To obtain a stable learning signal, we aggregate semantic gradients across the batch:

$$\bar{g}_\omega = \text{Aggregate}\left( \{g_i\}_{i=1}^B \right),$$

where $\text{Aggregate}(\cdot)$ denotes an LLM-based consolidation procedure that extracts recurring failure patterns and consistent refinement suggestions across trajectories, while filtering out conflicting or trajectory-specific signals. The resulting $\bar{g}_\omega = (\bar{g}^{(\mathcal{I})}, \bar{g}^{(\pi)}, \bar{g}^{(\beta)})$ represents a batch-averaged semantic gradient that captures systematic weaknesses of Skill $\omega$ revealed by experience.

**Semantic Skill Update.** We update Skill $\omega$ using the batch-averaged semantic gradient to obtain a *candidate Skill* $\omega'$:

$$\omega' = \omega \oplus \bar{g}_\omega,$$

where $\oplus$ denotes an LLM-driven update operation that revises $\mathcal{I}_\omega$, $\pi_\omega$, and $\beta_\omega$ according to the refinement directions encoded in $\bar{g}_\omega$, while preserving the overall Skill structure. This operation plays the role of a **gradient ascent step** in Non-Parametric PPO: *instead of updating numerical parameters, the Skill is updated along a direction intended to improve expected return, as suggested by aggregated hindsight feedback.*

## 4.2. PPO-Style Trust-Region Verification

Semantic-gradient updates are generated by an LLM from hindsight feedback. As a result, they may extrapolate be-

yond the observed interaction data and introduce hallucinated or behaviorally unstable Skills. To mitigate this risk, we introduce a PPO-style trust-region verification step to evaluate each candidate Skill before adding it to the pool.

We treat the frozen LLM as the underlying stochastic policy and evaluate a candidate Skill $\omega'$ using batch-size trajectories collected under the previous Skill $\omega$. For each timestep $t$, we compute an importance ratio

$$\rho_t(\omega') = \frac{\pi_{\text{LLM}}(a_t \mid s_t, \omega')}{\pi_{\text{LLM}}(a_t \mid s_t, \omega)}, \tag{6}$$

which measures how the likelihood of the historical action $a_t$ would change if the candidate Skill $\omega'$ were applied instead of the behavior Skill $\omega$ at the same state $s_t$. Since we do not train a value function, we estimate advantages using return-to-go with a running baseline:

$$G_t = \sum_{k=t}^{T-1} \gamma^{k-t} r_k, \quad \hat{A}_t = G_t - \bar{R},$$

where $\bar{R}$ is a running baseline used to reduce variance. We compute a PPO-style clipped surrogate *verification functional*, hereafter referred to as the **PPO Gate**, to evaluate the counterfactual advantage of applying the candidate Skill $\omega'$ on historical trajectories:

$$L^{\text{CLIP}}(\omega') =$$

$$\hat{\mathbb{E}}_{\tau \sim \mathcal{B}} \left[ \frac{1}{|\tau|} \sum_{t \in \tau} \min\left( \rho_t(\omega')\hat{A}_t,\ \text{clip}(\rho_t(\omega'), 1-\epsilon, 1+\epsilon)\,\hat{A}_t \right) \right].$$

This verification functional favors candidate Skills that assign higher probability to high-advantage actions observed in past trajectories, while limiting large deviations from the behavior policy, thereby enforcing a trust-region constraint.

**Best-of-$N_c$ Selection.** Given $N_c$ candidates generated via semantic gradients, we compute the PPO Gate score $J(\omega') \triangleq L^{\text{CLIP}}(\omega')$ for each and select the best candidate:

$$\omega_{\text{new}} = \arg\max_{\omega'} J(\omega'), \quad \text{subject to} \quad J(\omega_{\text{new}}) > 0.$$

Since the PPO Gate is based on advantage estimates, a positive score indicates that the candidate is expected to outperform the previous Skill under the trust-region constraint, filtering out unreliable or hallucinated candidates to prevent destabilizing shifts.

## 4.3. Score-Based Skill Pool Maintenance

Since the Skill pool has a fixed capacity $K$, indiscriminately adding new Skills increases storage and selection overhead. Beyond PPO Gate, we retain or prune Skills based on their empirical contribution, measured by an online score. During interaction, multiple Skills may be invoked within a trajectory, and rewards may be provided at different granularities. We therefore define a unified, advantage-style Skill gain. Given a trajectory $\tau$ with rewards $\{r_t\}$, we first form a

per-step advantage signal $\tilde{r}_t \triangleq r_t - \bar{r}$, where $\bar{r}$ is a running baseline. The gain of Skill $\omega$ in $\tau$ is defined as the average advantage accumulated over the time steps during which $\omega$ is active:

$$G(\omega; \tau) = \frac{1}{|\mathcal{T}_\omega(\tau)|} \sum_{t \in \mathcal{T}_\omega(\tau)} \tilde{r}_t, \tag{7}$$

where $\mathcal{T}_\omega(\tau)$ denotes the set of time steps when Skill $\omega$ is executed. When only a trajectory-level return $R(\tau)$ is available, we set $\tilde{r}_t \equiv (R(\tau) - \bar{R})/|\tau|$, yielding the same advantage-style definition.

**Online score update.** For each Skill $\omega$, we maintain a cumulative gain $G_b(\omega)$ and an invocation count $N_b(\omega)$. After processing a batch $\mathcal{T}^{(b)}$, we update the online score:

$$G_{b+1} = G_b + \sum G(\omega; \tau), \quad N_{b+1} = N_b + \sum c(\omega; \tau),$$
$$\text{Score}_{b+1} = \frac{G_{b+1}}{\max(1, N_{b+1})}.$$

**Online Score-Based Pruning.** To enforce the fixed pool capacity, we maintain the Skill pool using online scores. Specifically, we remove (i) Skills with non-positive online score, i.e., $\text{Score}(\omega) \leq 0$, which indicates no expected advantage over existing Skills, and (ii) duplicate or semantically redundant Skills. If the pool still exceeds capacity, we further prune Skills in ascending order of online score. As the baseline improves over time, this rule imposes evolutionary pressure, phasing out obsolete Skills while retaining those with consistently positive gains.

## 5. Experiments

To examine whether Skill-Pro learns reusable procedural memory (instantiated as Skills) from interaction experience, we conduct the following experiments:

- **RQ1:** Is the learned procedural memory efficiently reusable? (§ 5.2)
- **RQ2:** How do different components contribute to learning reusable skills? (§ 5.3)
- **RQ3:** How does procedural memory evolve and get reused in practice? (§ 5.4)

### 5.1. Experimental Setup

**Benchmarks.** We conduct experiments on **ALF-World** (Shridhar et al., 2021) and **TextArena** (Guertler et al., 2025), two canonical benchmarks for multi-turn sequential decision-making. ALFWorld separates training from out-of-distribution environments, while **Mastermind-v0** from TextArena spans three difficulty tiers; both supporting evaluation of cross-task memory reuse (see Appendix B.1). To further assess generalizability beyond text-based games and embodied environments, we additionally evaluate on the **Berkeley Function Calling Leaderboard (BFCL v4)** (Patil et al., 2025); results are reported in Appendix C.2.

**Baselines.** We compare Skill-Pro against a diverse set of memory-augmented and reasoning-based LLM agents. Memory-augmented baselines span raw trajectory retrieval (**RAG** (Lewis et al., 2020)), distilled insights (**Expel** (Zhao et al., 2024)), concise notes (**A-MEM** (Xu et al., 2025)), structured workflows (**AWM** (Wang et al., 2024)), and hybrid memory representations (**G-Memory** (Zhang et al., 2025a)). We further include representative reasoning-based baselines, including **ReAct** (Yao et al., 2022) and **CoT** (Wei et al., 2022), as well as a **State**-based agent without external memory. All methods use the same frozen LLM, ensuring a fair comparison without parameter fine-tuning.

**LLM Backbones.** We evaluate Skill-Pro with multiple LLM backbones. On TextArena, we learn Skills using `Gemma-2-9B` and evaluate their reuse across heterogeneous agents, including `Gemma-3-4B`, `Qwen3-32B`, and `LLaMA-3.3-70B-Instruct`. On ALFWorld, all experiments are conducted with `Qwen3-32B`.

**Evaluation Metrics.** We evaluate all methods along three complementary dimensions: **(1) Memory reuse** is measured by *in-domain, cross-task, and cross-agent reuse rates*, indicating probability that stored memory is reused per episode. **(2) Performance** is measured by an agent's episodic return. **(3) Efficiency** evaluates the *storage cost* and *inference cost* of memory reuse, measured by total tokens stored in memory (Total Stored Tokens), average tokens per memory units (Avg Tokens per Unit), additional prompt tokens added to the decision prompt ($\Delta$ Prompt Tokens/Step), and the probability of retrieving memory at each step (Retrieval Ratio). Metric details are provided in Appendix B.4.

### 5.2. Does Skill-Pro Truly Enable Reusability?

We evaluate Skill-Pro and baseline memory methods on memory reuse and efficiency (Table 1), as well as task performance (Table 2). Memories are built on in-domain tasks (Mastermind-v0 and ALFWorld-Train) and reused on out-of-distribution or higher-difficulty tasks (cross-task), and across agents with different LLM backbones (cross-agent); results are averaged over 50 episodes per setting.

**Skill-Pro's superior reuse rates validate both Skill-MDP effectiveness and learned Skills' quality.** As shown in Table 1, Skill-Pro's reuse rate consistently outperforms all baselines in in-domain, cross-task, and cross-agent evaluations. While baselines suffer from low reuse due to redundant episodic data, Skill-Pro's high reuse rate demonstrates that learned procedural Skills are both high-quality and inherently generalizable.

**Skill-Pro's high efficiency is evidenced by minimal storage and low execution overhead.** While baselines accumulate hundreds of thousands of tokens by storing diverse episodic units, such as trajectories, insights, and workflows, Skill-Pro maintains only **816 tokens**, demonstrating that pro-

*Table 1.* **Main Results on Memory Reuse and Efficiency.** Results are reported as Mean$_{\pm \text{Std Dev}}$. Unit types in "Avg Tokens Per Unit" denote T: Trajectory, I: Insights, N: Notes, W: Workflow, PT: Part of Trajectory, and Skill (ours). ↑ (↓) indicates higher (lower) is better.

| Method | Experience Reuse Rate (↑) | | | | | Efficiency Metrics (↓) | | | |
| --- | --- | --- | --- | --- | --- | --- | --- | --- | --- |
| | In-domain | Cross-task Reuse | | Cross-agent Reuse | | Storage Cost | | Execution Cost | |
| | Mastermind -v0 | Mastermind -v0-Hard | Mastermind -v0-Extreme | Gemma-3 -4B | Qwen3 -32B | Total Stored Tokens | Avg Tokens per Unit | Δ Prompt Tokens/Step | Retrieval Ratio |
| RAG | $0.349_{\pm 0.145}$ | $0.441_{\pm 0.002}$ | $0.467_{\pm 0.050}$ | $0.111_{\pm 0.}$ | $0.146_{\pm 0.064}$ | 116527 | $2675_{\pm 414}$ (T) | $2698_{\pm 414}$ | $1_{\pm 0.}$ |
| Expel | $0.285_{\pm 0.015}$ | $0.242_{\pm 0.024}$ | $0.258_{\pm 0.016}$ | $0.254_{\pm 0.013}$ | $0.270_{\pm 0.017}$ | 294447 | $642_{\pm 0}$ (I) $4568_{\pm 2541}$ (T) | $5210_{\pm 2541}$ | $1_{\pm 0.}$ |
| A-MEM | $0.020_{\pm 0.005}$ | $0.017_{\pm 0.002}$ | $0.015_{\pm 0.002}$ | $0.020_{\pm 0.003}$ | $0.018_{\pm 0.003}$ | 200129 | $1210_{\pm 3}$ (N) | $1214_{\pm 3}$ | $1_{\pm 0.}$ |
| AWM | $0.080_{\pm 0.010}$ | $0.063_{\pm 0.006}$ | $0.075_{\pm 0.007}$ | $0.073_{\pm 0.006}$ | $0.060_{\pm 0.010}$ | 391706 | $602_{\pm 0}$ (W) $2914_{\pm 297}$ (T) | $3658_{\pm 21}$ | $0.049_{\pm 0.009}$ |
| G-Memory | $0.091_{\pm 0.027}$ | $0.170_{\pm 0.063}$ | $0.092_{\pm 0.016}$ | $0.360_{\pm 0.162}$ | $0.264_{\pm 0.104}$ | 40510 | $100_{\pm 79}$ (I) $334_{\pm 2}$ (PT) | $434_{\pm 79}$ | $0.097_{\pm 0.027}$ |
| **Skill-Pro (Ours)** | $\mathbf{0.925}_{\pm 0.061}$ | $\mathbf{0.825}_{\pm 0.061}$ | $\mathbf{0.900}_{\pm 0.094}$ | $\mathbf{0.850}_{\pm 0.094}$ | $\mathbf{0.875}_{\pm 0.112}$ | **816** | $\mathbf{102}_{\pm 0}$ (Skill) | $273_{\pm 5}$ | $0.591_{\pm 0.016}$ |

*Table 2.* **Main Performance Results.** Results are Mean$_{\pm \text{Std Dev}}$; **bold** denotes best. Shading blue is normalized per column, indicating relative performance within each task. Both sections evaluate the performance of memory reuse or reasoning baselines: the **left** across **cross-task** difficulties with a fixed backbone; the **right** across **cross-agent** LLM backbones.

| Algorithm | ALFWorld (Success Rate ↑) | | Mastermind (Avg Return ↑) | | | Cross-agent (Mastermind-v0) | | |
| --- | --- | --- | --- | --- | --- | --- | --- | --- |
| | **Train** | **OOD** | **v0** | **Hard** | **Extreme** | Gemma-3 4B-it | Qwen3 32B | Llama-3.3 70B |
| **State** | $0.312_{\pm 0.040}$ | $0.262_{\pm 0.062}$ | $0.388_{\pm 0.236}$ | $0.336_{\pm 0.183}$ | $0.272_{\pm 0.129}$ | $0.414_{\pm 0.101}$ | $0.497_{\pm 0.159}$ | $0.613_{\pm 0.201}$ |
| RAG | $0.480_{\pm 0.134}$ | $0.402_{\pm 0.264}$ | $0.521_{\pm 0.236}$ | $0.344_{\pm 0.159}$ | $0.241_{\pm 0.136}$ | $0.404_{\pm 0.191}$ | $0.558_{\pm 0.204}$ | $0.620_{\pm 0.211}$ |
| CoT | $0.600_{\pm 0.069}$ | $0.620_{\pm 0.068}$ | $0.531_{\pm 0.063}$ | $0.381_{\pm 0.043}$ | $0.254_{\pm 0.031}$ | $0.417_{\pm 0.120}$ | $0.470_{\pm 0.153}$ | $0.542_{\pm 0.206}$ |
| ReAct | $0.580_{\pm 0.070}$ | $0.640_{\pm 0.068}$ | $0.557_{\pm 0.059}$ | $0.405_{\pm 0.074}$ | $0.263_{\pm 0.048}$ | $0.408_{\pm 0.131}$ | $0.425_{\pm 0.125}$ | $0.604_{\pm 0.230}$ |
| Expel | $0.680_{\pm 0.065}$ | $0.740_{\pm 0.063}$ | $0.424_{\pm 0.033}$ | $0.305_{\pm 0.031}$ | $0.239_{\pm 0.024}$ | $0.429_{\pm 0.117}$ | $0.483_{\pm 0.185}$ | $0.575_{\pm 0.27}$ |
| A-MEM | $0.520_{\pm 0.071}$ | $0.640_{\pm 0.068}$ | $0.471_{\pm 0.042}$ | $0.310_{\pm 0.038}$ | $0.253_{\pm 0.026}$ | $0.388_{\pm 0.115}$ | $0.570_{\pm 0.230}$ | $0.542_{\pm 0.162}$ |
| AWM | $0.700_{\pm 0.065}$ | $0.900_{\pm 0.042}$ | $0.546_{\pm 0.052}$ | $0.299_{\pm 0.036}$ | $0.294_{\pm 0.040}$ | $0.417_{\pm 0.144}$ | $0.592_{\pm 0.183}$ | $0.550_{\pm 0.238}$ |
| G-Memory | $0.681_{\pm 0.079}$ | $0.812_{\pm 0.017}$ | $0.577_{\pm 0.052}$ | $0.406_{\pm 0.056}$ | $\mathbf{0.356}_{\pm 0.036}$ | $0.428_{\pm 0.039}$ | $0.475_{\pm 0.190}$ | $0.535_{\pm 0.079}$ |
| **Skill-Pro (Ours)** | $\mathbf{0.900}_{\pm 0.105}$ | $\mathbf{0.909}_{\pm 0.287}$ | $\mathbf{0.606}_{\pm 0.234}$ | $\mathbf{0.463}_{\pm 0.210}$ | $0.333_{\pm 0.118}$ | $\mathbf{0.444}_{\pm 0.161}$ | $\mathbf{0.615}_{\pm 0.290}$ | $\mathbf{0.647}_{\pm 0.236}$ |

cedural memory is far more compact than episodic memory. Skill-Pro's lean representation, as reflected by Δ Prompt Tokens/Step, prevents prompt bloat and minimizes LLM execution load. Furthermore, by utilizing temporally extended "Skills", Skill-Pro reduces per-step retrieval ratio, ensuring highly efficient agent execution.

**Skill-Pro achieves superior performance despite a highly compressed memory footprint.** As shown in Table 2, our learned memory consistently yields performance gains when reused across varying task difficulties and LLM backbones of different scales. Notably, even under extreme memory compression, Skill-Pro maintains the highest success rates, reaching **0.90** in ALFWorld. This superior performance confirms that our framework successfully captures essential task logic, ensuring that only high-quality, decision-critical content is stored.

**5.3. Why Does Skill-Pro Work?**

To evaluate the contribution of each component, we conduct an ablation study comparing Skill-Pro against several variants, primarily using the Mastermind-v0 environment in TextArena. Beyond performance and reuse rate, we introduce two metrics: **Online Score** (average Skill quality in the pool) and **PPO Gate Pass Rate** (the ratio of candidates satisfying PPO Gate).

- **w/o Skill**: Utilizes only states for decision-making.
- **w/o NP-PPO**: Employs fixed skill seeds without the NP-PPO evolution process.
- **w/o SG**: Replaces Semantic Gradients with trajectory summaries; directly utilizing raw trajectories would otherwise trigger context window overflow for Gemma-2-9B.
- **w/o PPO Gate**: Removes the PPO Gate, allowing all generated candidates to enter the skill pool unconditionally.
- **w/o Score (FIFO)**: Replaces score-based pruning with a First-In-First-Out (FIFO) to manage pool capacity.

**Both the procedural Skill and NP-PPO evolution are fundamental to task success.** As shown in Table 3, the **w/o Skill** variant suffers a sharp performance drop (0.606 → 0.388), confirming that skills are essential building blocks for complex decision-making. While initial seeds provide a functional baseline, the **w/o NP-PPO** results highlight that our evolution mechanism is critical for refining general seeds into task-specific expertise, significantly boosting both reuse and success rates.

**Semantic Gradients and the PPO Gate are indispensable for the generation and verification of high-quality Skills.** Ablating either component degrades pool quality. Specifically, **w/o SG** triggers a 30% drop in the PPO Gate Pass Rate, proving that Semantic Gradients significantly enhance

*Table 3.* **Ablation Study on Skill-Pro.** Subscripts show relative change to the Full version (blue for degradation).

| Methods | Reuse Rate (↑) | Perfor-mance (↑) | Online Score (↑) | PPO Gate Pass Rate (↑) |
|---|---|---|---|---|
| **Skill-Pro (Full)** | **0.925** | **0.606** | **0.0406** | **59.49%** |
| w/o Skill | N/A | 0.388 (-36.0%) | N/A | N/A |
| w/o NP-PPO | 0.563 (-39.1%) | 0.482 (-20.5%) | 0.0265 (-34.7%) | N/A |
| *Ablation on NP-PPO* | | | | |
| w/o SG | 0.306 (-66.9%) | 0.530 (-12.5%) | 0.0015 (-96.3%) | 41.54% (-30.2%) |
| w/o PPO Gate | 0.222 (-76.0%) | 0.453 (-25.2%) | 0.0011 (-97.3%) | 100.00% (+68.1%) |
| w/o Score (FIFO) | 0.131 (-85.8%) | 0.439 (-27.6%) | -0.0064 (-115.8%) | 57.18% (-3.9%) |

Standard deviations are provided in Table 4 in Appendix.

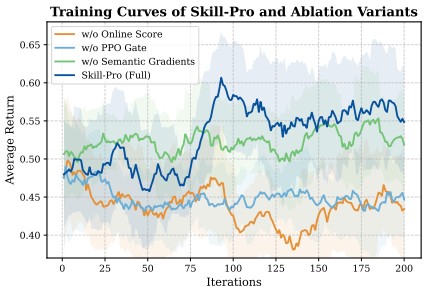

*Figure 3.* **Training curves of Skill-Pro and ablation variants.** Solid lines and shaded areas denote the smoothed mean and standard deviation of average returns, respectively.

the quality of generated skill candidates. Conversely, **w/o PPO Gate** admits all candidates without verification, destabilizing training as evidenced in the training curves (Fig. 3). Notably, **w/o SG** remains more stable than **w/o PPO Gate**, as its candidates must still pass trust-region verification.

**Score-based maintenance is critical for preserving evolutionary gains within the skill pool.** The **w/o Score (FIFO)** variant exhibits the most severe degradation among all NP-PPO ablations. Despite maintaining a high PPO Gate Pass Rate, FIFO inadvertently replaces high-performing Skills with unproven newcomers. The resulting negative Online Score ($-0.0018$) confirms that without score-based pruning, the pool fails to retain superior procedural knowledge, ultimately leading to a collapse in long-term performance.

### 5.4. How Does Skill-Pro Evolve and Reuse?

**Cross-Agent and Cross-Task Generalization.** Fig. 5 characterizes skill reuse through invocation probability and mean frequency per episode ($N$). Our analysis reveals that while different LLM backbones exhibit distinct usage profiles, the underlying selection patterns remain remarkably stable across varying task difficulties. Specifically, in cross-agent evaluations, `Gemma2-9B` shows a heightened reliance on `FBInference`, whereas `StratPlan` maintains consistent activation levels across agents, acting as a standardized procedural primitive. Crucially, in cross-task evaluations on *Mastermind-v0*, the skill distribution remains invariant as difficulty scales from Base to Extreme. This consistency

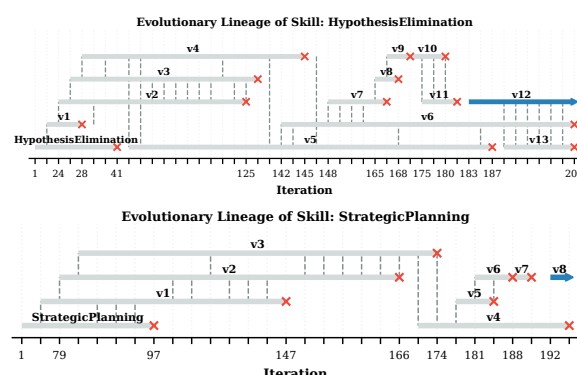

*Figure 4.* **Evolutionary Lineage of Skills.** Gray bars represent Skill lifespans along the evolutionary timeline (horizontal axis). Dashed vertical lines denote refinement events where a parent Skill evolves into children; multiple lines indicate repeated refinements. Red 'X' markers signify pruning of underperforming variants for pool efficiency. The dark blue arrow and sequential alignment (e.g., $v_1 \rightarrow v_{13}$) track the sustained trajectory of Skill evolution.

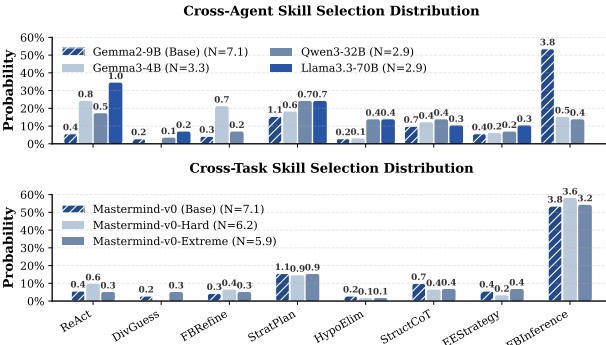

*Figure 5.* **Skill distribution across LLM agents and task complexities.** Bars represent the empirical invocation probability for each skill categorized by different LLM backbones (top) and task difficulty levels in *Mastermind-v0* (bottom). $N$ denotes the average number of skill invocations per episode.

demonstrates that Skill-Pro effectively distills the fundamental task logic, enabling robust generalization across environment complexities.

**Evolutionary Dynamics.** Fig. 4 illustrates the evolutionary trajectory of two representative Skills. The observed refinements and score-based pruning events underscore Skill-Pro's ability to evolve a compact, high-utility skill pool. Detailed analysis is provided in Appendix C.1.

## 6. Conclusion

We presented Skill-Pro, a framework enabling LLM agents to **autonomously learn procedural Skills** without parameter updates. By formalizing the **Skill-MDP**, Skill-Pro transforms passive episodic narratives into executable, reusable Skills, eliminating redundant on-the-fly reasoning. To ensure reliability without capability degradation, we propose **Non-Parametric PPO**, which leverages semantic gradients for high-quality candidate generation and a PPO Gate for

robust Skill verification. Finally, a score-based maintenance mechanism prunes low-return Skills to sustain long-term memory quality.

Results across diverse scenarios confirm that Skill-Pro achieves **superior reuse rates and significant performance gains with extreme memory compression**. Our findings validate that **high-quality procedural memory is fundamentally more efficient than raw episodic storage** for long-term autonomy. Future work will integrate implicit execution modules to better emulate human-like intelligence. Ultimately, the autonomous accumulation of procedural expertise via interaction without parameter updates represents a pivotal milestone toward the emergence of truly self-evolving artificial intelligence.

## Impact Statement

This paper presents work whose goal is to advance the field of machine learning by improving the efficiency and reusability of autonomous agents. By enabling procedural knowledge accumulation without continuous parameter updates, our work contributes to more resource-efficient and sustainable AI development. There are many potential societal consequences of our work, none of which we feel must be specifically highlighted here.

## Acknowledgments

H. Zhang is supported by the National Natural Science Foundation of China (No. 72450002). H. Li is supported by the Beijing Major Science and Technology Project (No. Z251100008425006) and the Beijing Natural Science Foundation (No. L257007). Y. Wang is supported by the Beijing Major Science and Technology Project (No. Z251100008425006), the National Natural Science Foundation of China (Nos. 92370129 and 62376010), the Beijing Natural Science Foundation (No. L257007), and the Beijing Nova Program (Nos. 20230484344 and 20240484642).

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

---

**Algorithm 1** Non-Parametric PPO for Skill Evolution

---

**Input:** Initial Skill pool $\Omega_0$, Frozen LLM $\pi_{\text{LLM}}$, Capacity $K$

Initialize online scores $\text{Score}(\omega) = 0$ for all $\omega \in \Omega_0$

**for** $n = 1$ **to** $N$ **do**

    // **1. Experience Collection**

    Collect a batch of trajectories $\mathcal{T}^{(B)}$ using policy $\pi_\Omega = \mu \cdot \pi_{\text{LLM}}$.

    // **2. Semantic Gradient Extraction & Optimization**

    **for** each Skill $\omega \in \Omega_n$ invoked in $\mathcal{T}^{(B)}$ **do**

        Extract per-trajectory semantic gradients $\{g_i\}$ via hindsight attribution.

        $\bar{g}_\omega = \text{Aggregate}(\{g_i\}_{i=1}^{B})$ {Batch-level aggregation}

        Generate $N_c$ candidates $\{\omega'_j\}$ where $\omega' = \omega \oplus \bar{g}_\omega$ via LLM.

        // **3. PPO-Style Trust-Region Verification (PPO Gate)**

        **for** each candidate $\omega'_j$ **do**

            Compute $J(\omega'_j) = \hat{\mathbb{E}}[\min(\rho_t \hat{A}_t, \text{clip}(\rho_t, 1 - \epsilon, 1 + \epsilon)\hat{A}_t)]$

        **end for**

        $\omega^* = \arg\max_{\omega'_j} J(\omega'_j)$

        **if** $J(\omega^*) > 0$ **then**

            $\Omega = \Omega \cup \{\omega^*\}$

        **end if**

    **end for**

    // **4. Skill Pool Maintenance**

    Update $\text{Score}(\omega)$ based on cumulative gain $G(\omega; \tau)$.

    **if** $|\Omega| > K$ **then**

        Prune Skills with $\text{Score}(\omega) \leq 0$ or those with lowest scores, semantically redundant items via cosine similarity.

    **end if**

**end for**

**Output:** Optimized Skill pool $\Omega_N$

---

## Summary of Appendices

---

## A. Full Related Works

**Learning from Interaction Experience in LLM Agents.** Recent LLM-agent frameworks improve sequential decision making by leveraging interaction experience, either through *parametric fine-tuning* or via *non-parametric* adaptation at inference time. Parametric methods, such as reinforcement learning (RL), incorporate feedback from interaction by updating model parameters and have demonstrated strong task performance (Ouyang et al., 2022; Rafailov et al., 2023; Guo et al., 2025). However, as pretrained LLMs become increasingly capable general-purpose reasoners, task-specific fine-tuning is often *computationally expensive*, tends to *over-specialize models to narrow task distributions*, and can *degrade general-purpose behaviors* under continual adaptation (Ziegler et al., 2020; Shi et al., 2025a; Luo et al., 2025). These limitations have motivated growing interest in *non-parametric* approaches that learn from interaction experience *without*

*updating model parameters*, among which memory-augmented LLM agents constitute a dominant paradigm (Zhao et al., 2024; Zhou et al., 2025a).

**Memory-Augmented LLM Agents.** Memory-augmented LLM agents store past interaction experience in an external memory and retrieve relevant content to condition the LLM's reasoning during decision making, thereby extending the agent's effective temporal horizon without updating model parameters (Zhao et al., 2024; Yang et al., 2025; Cai et al., 2025). Existing methods mainly differ in what experience is stored, how it is retrieved, and how the memory is updated over time. The most basic form stores **raw trajectories or episodic records** and retrieves full or partial past episodes to guide current decisions in a case-based manner (Park et al., 2023). To improve efficiency and generalization, several approaches distill experience into **abstract summaries** (Yang et al., 2025), **high-level principles** (Wu et al., 2025; Agrawal et al., 2025; Cai et al., 2025), or **insights** extracted from past successes or failures (Zhao et al., 2024). To capture complex dependencies across long horizons, **structured or graph-based memory** organizes experience into hierarchical or graph representations, such as G-Memory (Zhang et al., 2025a) and HippocRAG (Jimenez Gutierrez et al., 2024), enabling multi-hop retrieval and reasoning (Rezazadeh et al., 2024; Xia et al., 2025). In parallel, **dense vector compression** encodes experience into latent embeddings or matrices to support scalable storage and similarity-based retrieval, as in LARIMAR (Das et al., 2024) and MemGen (Zhang et al., 2025b). More recent work maintains **dynamic knowledge snippets**, such as textual notes or discrete knowledge units inspired by human note-taking, which are continuously updated during interaction, including Self-RAG (Asai et al., 2024), ReMemR1 (Shi et al., 2025b), MemGen (Zhang et al., 2025b), and Mem1 (Zhou et al., 2025b). Finally, some approaches store explicit task-completion paths or workflows that can be retrieved to guide future actions (Wang et al., 2024). Despite these advances, existing memory-augmented agents prioritize experience storage over content reusability. As interaction trajectories grow, this paradigm inevitably accumulates massive redundancy, leading to prohibitive storage and retrieval overhead. Furthermore, treating retrieved episodes as passive context forces agents to repetitively re-reason actions within limited context windows, imposing significant inference pressure.

Relatedly, skill-based agents (Wang et al., 2023; Tan et al., 2024) and procedural knowledge acquisition (Zhu et al., 2023; Sumers et al., 2023) explore capturing executable behaviors. Recent studies (Han et al., 2025; Fang et al., 2025) have pioneered procedural memory mechanisms, yet optimizing execution reusability remains an open problem. To bridge this gap, we propose Skill-Pro to formalize and learn reusable procedural memory from interaction experience, ensuring efficient and reliable long-term autonomy.

# B. Detailed Experimental Setup

## B.1. Benchmarks

We evaluate Skill-Pro on **TextArena** (Guertler et al., 2025) and **ALFWorld** (Shridhar et al., 2021), two benchmarks that capture the core challenges of experience reuse in sequential decision-making. TextArena consists of multi-turn, text-based games with varying levels of difficulty. These tasks require long-horizon reasoning and adaptation to iterative feedback, making them well suited for evaluating repeated reuse of accumulated experience both within and across tasks. ALFWorld is an embodied environment grounded in natural language, involving long action sequences and high-level decision-making in an abstract state space. Importantly, ALFWorld explicitly separates training tasks from out-of-distribution evaluation tasks, enabling direct assessment of experience reuse under distribution shift.

## B.2. Baselines

We compare Skill-Pro against a comprehensive set of memory-augmented and reasoning-based LLM agents. Memory-augmented baselines differ in how experience is stored, including raw interaction trajectories (**RAG**) (Lewis et al., 2020), distilled insights (**Expel**) (Zhao et al., 2024), concise notes (**A-MEM**) (Xu et al., 2025), structured workflows (**AWM**) (Wang et al., 2024), and hybrid memory representations (**G-Memory**) (Zhang et al., 2025a). All of these methods retrieve past experience to condition decision-making. We additionally include representative reasoning-based baselines, including **ReAct** (Yao et al., 2022) with chain-of-thought (**CoT**) (Wei et al., 2022) reasoning, as well as a minimal **State**-based agent that directly selects actions from the current environment state without external memory. All methods use the same frozen LLM for decision-making, with no parameter fine-tuning, ensuring a fair and controlled comparison.

### B.3. LLM Backbones

To evaluate robustness across model scales and architectures, we conduct experiments with multiple LLM backbones. On TextArena, the main experiments are performed using `Gemma-2-9B`, and the resulting Skill pool is reused across heterogeneous LLM agents, including `Gemma-3-4B`, `Qwen3-32B`, and `LLaMA-3.3-70B-Instruct`, to assess cross-agent reuse efficiency and performance. On ALFWorld, all experiments are conducted using `Qwen3-32B`.

### B.4. Evaluation Metrics

We evaluate all methods along three complementary dimensions: *task performance*, *memory reuse*, and *efficiency*. **(1) Task performance** measures an agent's ability to solve sequential decision-making tasks. **(2) Experience reuse** is quantified by **in-domain, cross-task, and cross-agent reuse rates**, capturing how effectively stored memory is reused across tasks and LLM backbones. **(3) Efficiency** measures the cost of memory reuse, including storage footprint and inference-time overhead, quantified by total stored tokens, average tokens per memory unit, retrieval ratio, and additional prompt tokens per step.

**(1) Task Performance.** We report standard benchmark-specific performance metrics. For TextArena environments, we use the average return per episode. For ALFWorld, we report success rate, which directly reflects task completion performance.

**(2) Reuse Metrics.** To quantify how effectively stored experience is reused, we introduce three reuse metrics. Let $\mathcal{M}$ denote the set of stored units.

**In-domain Reuse Rate ($\uparrow$).** This metric measures how much stored experience is actually reused within the same task domain. It is defined as the fraction of stored units that are invoked at least once during evaluation:

$$\text{In-domain Reuse Rate} = \frac{|\{u \in \mathcal{M} \mid \text{used}(u) \geq 1\}|}{|\mathcal{M}|}.$$

**Cross-task Reuse Rate ($\uparrow$).** This metric evaluates whether experience generalizes across tasks. It is defined as the fraction of stored units that are reused in target tasks different from those in which they were learned:

$$\text{Cross-task Reuse Rate} = \frac{|\{u \in \mathcal{M} \mid \exists \tau \in \mathcal{T}_{\text{target}}, \ \text{used}_\tau(u) \geq 1\}|}{|\mathcal{M}|}.$$

**Cross-agent Reuse Rate ($\uparrow$).** This metric measures whether stored experience can be reused across different LLM backbones. It is defined as the fraction of stored units that are reused by at least one alternative LLM agent:

$$\text{Cross-agent Reuse Rate} = \frac{|\{u \in \mathcal{M} \mid \exists a \in \mathcal{A}, \ \text{used}_a(u) \geq 1\}|}{|\mathcal{M}|},$$

where $\mathcal{A}$ denotes the set of evaluated LLM agents.

**(3) Efficiency Metrics.** Beyond reuse effectiveness, we measure the efficiency of experience storage and reuse, including both memory footprint and inference-time overhead.

**Total Stored Tokens ($\downarrow$).** This metric quantifies the overall storage footprint by summing the token counts of all stored units:

$$\text{Total Stored Tokens} = \sum_{u \in \mathcal{M}} \text{tokens}(u).$$

**Avg Tokens per Unit ($\downarrow$).** This metric measures representation compactness and is defined as the average token length per stored unit:

$$\text{Avg Tokens per Unit} = \frac{1}{|\mathcal{M}|} \sum_{u \in \mathcal{M}} \text{tokens}(u).$$

**Retrieval Ratio ($\downarrow$).** This metric measures how frequently experience reuse is triggered during decision-making. It is defined as the fraction of decision steps in which a stored unit is retrieved or activated:

$$\text{Retrieval Ratio} = \frac{\sum_{t=1}^{T} \mathbb{I}[\text{reuse}_t]}{T},$$

*Table 4.* **Ablation Study on Skill-Pro Components.**

| Methods | Reuse Rate (↑) | Perfor-mance (↑) | Online Score (↑) | PPO Gate Pass Rate (↑) |
|---|---|---|---|---|
| **Skill-Pro (Full)** | **0.925**$_{\pm 0.061}$ | **0.606**$_{\pm 0.234}$ | **0.0406**$_{\pm 0.0022}$ | **59.49%**$_{\pm 49.09\%}$ |
| w/o Skill | – | 0.388$_{\pm 0.236}$ | – | – |
| w/o NP-PPO | 0.563$_{\pm 0.176}$ | 0.482$_{\pm 0.197}$ | 0.0265$_{\pm 0.0270}$ | – |
| *Ablation on NP-PPO* | | | | |
| w/o SG | 0.306$_{\pm 0.070}$ | 0.530$_{\pm 0.184}$ | 0.0015$_{\pm 0.0003}$ | 41.54%$_{\pm 36.06\%}$ |
| w/o PPO Gate | 0.222$_{\pm 0.083}$ | 0.453$_{\pm 0.167}$ | 0.0011$_{\pm 0.0033}$ | 100.00%$_{\pm 0.00\%}$ |
| w/o Score (FIFO) | 0.131$_{\pm 0.052}$ | 0.439$_{\pm 0.186}$ | -0.0064$_{\pm 0.0052}$ | 57.18%$_{\pm 42.06\%}$ |

*– indicates Not Applicable.*

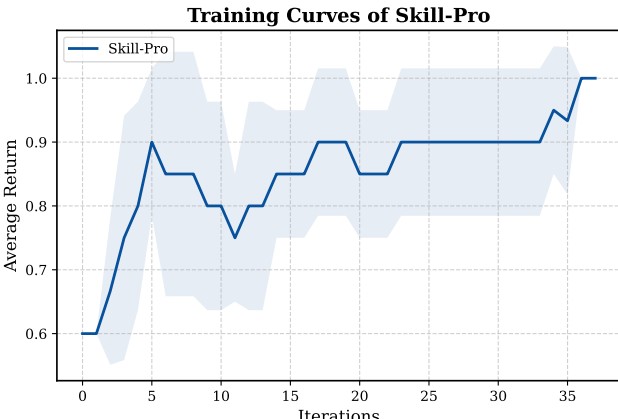

*Figure 6.* **Training curves of ALFWorld**. Solid lines and shaded areas denote the smoothed mean and standard deviation of average returns, respectively.

where $\mathbb{I}[\cdot]$ is an indicator function and $T$ is the total number of decision steps.

**ΔPrompt Tokens / Step (↓).** This metric captures the additional inference burden introduced by experience reuse. It is defined as the average increase in prompt tokens relative to a state-only prompt:

$$\Delta\text{Prompt Tokens / Step} = \frac{1}{T}\sum_{t=1}^{T}\Big(\text{tokens}(\text{prompt}_t) - \text{tokens}(\text{state}_t)\Big).$$

For comparable task performance, a lower value indicates reduced inference overhead and less reliance on large contextual inputs.

## C. Additional Experimental Details

### C.1. Evolutionary Lineage Analysis.

Fig. 4 visualizes the evolutionary lineage of Skills, offering a transparent view of how Skills are iteratively refined and consolidated within the Skill Pool. Multiple vertical dashed links between successive variants—such as $v_2$–$v_4$ of *HypothesisElimination*—indicate repeated refinement cycles in which several candidate variants are temporarily retained until a superior version is validated by online scores. The frequent appearance of red 'X' markers across both *HypothesisElimination* and *StrategicPlanning* highlights the critical role of **online score–based pruning** in maintaining Skill Pool efficiency, preventing uncontrolled accumulation of redundant variants that would otherwise degrade performance. Ultimately, each lineage converges to a persistent Skill (marked by the dark blue arrow), such as *HypothesisElimination* $v_{12}$, which remains stable after the exploration phase.

## C.2. Evaluation on Function Calling Tasks

To evaluate the generalizability of Skill-Pro beyond text-based games and embodied environments, we benchmark our framework on the Berkeley Function Calling Leaderboard (BFCL v4) (Patil et al., 2025), an authoritative benchmark for assessing LLM proficiency in tool invocation across diverse real-world applications. BFCL v4 encompasses challenging evaluation scenarios including single-turn and multi-turn function execution, parallel tool calls, and the detection of irrelevant or uncallable queries, allowing us to verify whether accumulated procedural Skills can govern precise, programmatic tool-use tasks.

**Baseline Selection.** We compare Skill-Pro against the three reasoning baselines used in our main experiments (State, ReAct, and CoT; see §B.2 for descriptions).

**Experimental Results.** Skill-Pro's learned procedural Skills generalizes effectively to function calling, achieving the highest accuracy across all baselines. As shown in Table 5, Skill-Pro attains an accuracy of **0.433**, outperforming ReAct (0.383) and CoT (0.367) by a substantial margin. This result demonstrates that the procedural Skills learned by Skill-Pro successfully capture reusable execution logic, such as structured parameter validation and systematic constraint checking, that transfers beyond the training environment, validating its robust generalizability to practical software tool-use scenarios.

*Table 5.* **Performance on the BFCL v4 function calling benchmark.** Accuracy is reported as Mean±Std Dev. Skill-Pro's procedural Skills provide consistent gains over reasoning-only baselines in tool-use scenarios.

| Method | Accuracy ($\uparrow$) |
|---|---|
| State | $0.233 \pm 0.025$ |
| ReAct | $0.383 \pm 0.023$ |
| CoT | $0.367 \pm 0.023$ |
| **Skill-Pro (Ours)** | $\mathbf{0.433 \pm 0.050}$ |

## D. Case Study

### D.1. Semantic Gradient Generation

This section illustrates the generation of semantic gradients within the Mastermind environment. By presenting a representative failure trajectory, we demonstrate the resulting **Semantic Gradient** and its corresponding **Trajectory Summary** used in our ablation studies.

The semantic gradient is structured as a tuple of updates for the Initiation ($I$), Policy ($\pi$), and Termination ($\beta$) components. Notably, if a component requires no adjustment, its gradient is represented as an **empty string** (`""`), ensuring that the evolution remains focused solely on identified errors. For comparison, we also provide the **Trajectory Summary** to highlight the distinction between neutral, fact-based compression and our proposed diagnostic gradients.

---

**Example: Old Skill and Execution Trajectory**

**Skill Definition: StrategicPlanning**

- **Initiation**: At the very beginning of the game (Turn 1) when no previous feedback exists.
- **Strategy Steps**:
    1. Choose a mathematically diverse starting guess (e.g., all unique).
    2. Establish the initial search boundaries based on allowed range.
    3. Set the logic for tracking historical moves.
- **Termination**: The first valid move is submitted and initial feedback is received.

---

**Execution Trajectory (Mastermind-v0):**

```
[GAME] You are playing Mastermind.
```

```
You need to find the code that is 4 digits long, each digit from 1 to 6,
with no duplicates.
In your response, you can submit your guess in the following format: '[2 1 4 5]'.
After each guess, you will receive feedback in the form of black and white pegs.
A black peg indicates a correct digit in the correct position, while a white peg
indicates a correct digit in the wrong position.
You have 20 turns to guess the code.

[Player] [1 2 3 4]
[GAME] Submitted [1 2 3 4]. Feedback: 0 black peg(s), 2 white peg(s).
[Player] [1 2 3 4]
[GAME] You attempted an invalid move. Reason: You have already guessed [1, 2, 3, 4].
Please try a different guess. Please resubmit a valid move and remember to follow
the game rules to avoid penalties.
[Player] [5 6 1 2]
[GAME] Submitted [5 6 1 2]. Feedback: 1 black peg(s), 2 white peg(s).
[Player] [3 6 2 4]
[GAME] Submitted [3 6 2 4]. Feedback: 1 black peg(s), 1 white peg(s).
[Player] [1 3 5 6]
[GAME] Submitted [1 3 5 6]. Feedback: 1 black peg(s), 3 white peg(s).
[Player] [1 2 3 4]
[GAME] You attempted an invalid move. Reason: You have already guessed [1, 2, 3, 4].
Please try a different guess. Please resubmit a valid move and remember to follow
the game rules to avoid penalties.
[Player] [2 3 5 6]
[GAME] Submitted [2 3 5 6]. Feedback: 1 black peg(s), 2 white peg(s).
[Player] [3 2 1 6]
[GAME] Submitted [3 2 1 6]. Feedback: 3 black peg(s), 0 white peg(s).
[Player] [3 2 1 6]
[GAME] You attempted an invalid move. Reason: You have already guessed [3, 2, 1, 6].
Please try a different guess. Please resubmit a valid move and remember to follow
the game rules to avoid penalties.
[Player] [3 2 1 6].
Game Over!
Final reward: 0.75.
Feedback: Invalid Move: You have already guessed [3, 2, 1, 6].
Please try a different guess.
```

**Generated Semantic Gradient (Raw Content)**

**diagnosis:**
*"The agent repeatedly re-submitted previously used guesses, violating the 'no duplicate guesses' rule. This indicates a missing (or not enforced) move-history tracking step, leading to invalid-move penalties and ending the episode despite having a strong near-solution state (e.g., 3 black pegs)."*
**is_related:** `true`

**semantic_gradient:**

**`initiation`** ""

**`policy`** "Add a strict 'guess history' ledger and enforce a pre-submit validity gate: before submitting any guess, normalize it (strip punctuation, ensure 4 digits, digits 1-6, no duplicates), then check it is NOT in prior_guesses. If it was used, generate the nearest alternative consistent with current constraints (e.g., keep confirmed positions, permute remaining digits, or swap two non-confirmed positions) and re-check until a new valid guess is found. Never repeat a guess even when backtracking; always consult the ledger first."

**`termination`** "Do not terminate after the first feedback. Terminate only after (a) you have received the initial feedback AND (b) you have initialized and stored: prior_guesses, digit-set constraints from

feedback, and a rule that forbids repeating any prior guess. If an invalid-move warning appears, immediately terminate the current skill and hand control back with an explicit note: 'Duplicate guess attempted; must choose a new unseen guess via history check.' "

---

**Example: Trajectory Summary (Ablation Baseline)**

**Summary Output:**
"The player played a game of Mastermind. The player had 20 turns to guess a 4-digit code. The player received feedback on their guesses in the form of black and white pegs. The player guessed the code on their 10th turn. The final reward was 0.75."

---

## D.2. How Skills Guide Primitive Actions

To illustrate how procedural memory guides primitive action generation during execution, we provide an example from the ALFWorld environment.

In this scenario, the agent arrives at `fridge 1` and receives the environment feedback `The fridge 1 is closed`, imposing a physical constraint that must be satisfied before any object can be retrieved. Without procedural memory, the agent ignores this constraint and directly attempts `take tomato 1 from fridge 1`, producing an invalid action. With the `SelfConsistencyCheck` Skill activated, the agent first validates the candidate action against the current state, detects the constraint violation, and revises the action to the valid prerequisite `open fridge 1`. This example shows how a learned Skill conditions the frozen LLM to generate constraint-aware primitive actions.

We note that this qualitative case illustrates the mechanism by which Skills guide primitive action selection, rather than demonstrating efficiency gains at individual steps. The value of procedural memory lies in its reusability across recurring situations, enabling Skills to improve overall task performance over extended horizons, as quantitatively validated in §5.2.

---

**Execution Example: Navigating Environmental Constraints**

**State Observation**
```
You are in the middle of a room. Looking quickly around you, you see a
microwave 1, a cabinet 3,...(remaining details omitted)

Your task is to: clean some tomato and put it on countertop.

> go to fridge 1
> You arrive at loc 7. The fridge 1 is closed.
```

**Activated Skill:** `SelfConsistencyCheck`

- **Initiation:** When the action must strictly satisfy known rules or historical constraints.

- **Procedure:**
    1. Propose a candidate action.
    2. Check whether this action violates any known rules or past feedback.
    3. If a violation is found, revise the action to remove the violation.
    4. Repeat the check until no violation remains.
    5. Output the final consistent action.

- **Termination:** An action that passes all self-consistency checks is produced.

**Output Action Comparison**

- **Without skill:** `take tomato 1 from fridge 1`  (*Invalid*)

- **With skill:** `open fridge 1`  (*Valid Prerequisite*)

---

# E. Prompt Template

---

**Prompt Template for Skill-based Decision Making**

**System Instruction:**
(If applicable) You are an embodied agent in a simulated house. Your goal is to complete a specific household task (e.g., put a clean sponge in the cabinet).
**Inputs:**

- **Current State**: {state}

- **Admissible Commands**: {admissible_commands} (Optional, e.g., for ALFWorld)

- **Active Skill**: {skill_text} (The strategy currently being executed)

**Instructions for Skill Usage:**

1. **Match**: Decide if the active skill's Target Situation fits the current state (Yes/No).

2. **Apply**: Execute EACH strategy step one by one. For each step, explicitly reference the relevant part of the CURRENT STATE or feedback history.

3. **Output**: You MUST output ONLY one action in the specified format.

**Environment-Specific Constraints:**

- **FrozenLake**: <action>[direction]</action>. Valid: [up], [down], [left], [right].

- **Mastermind**: <action>[d1 d2 d3 ...]</action>. Numbers only, no duplicates, never repeat a past guess.

- **Hangman**: <action>[letter]</action> or <action>[word]</action>. Guess to reduce uncertainty.

- **ALFWorld**: <action>...</action>. Content MUST be chosen from ADMISSIBLE COMMANDS.

**Response Requirements:** Your output must strictly follow this format:

```
<think>
match: Yes/No + short reason
apply:
    - Step 1: ...
    - Step 2: ...
</think>
<action>[Specific Action]</action>
```

---

**Prompt Template for Skill Termination**

You are a Meta-Controller supervising an AI Agent to determine if the current skill should be terminated.
**Inputs:**

- **Current State**: {state}

- **Active Skill**:
    - **Name**: {skill.name}
    - **Initiation (When to use)**: {skill.initiation}
    - **Termination (When to stop)**: {skill.termination}

**Instructions:** Decide whether the agent should **STOP** using this skill based on the following logic:

1. **Termination Met**: Return DONE if the Termination condition is already achieved in the CURRENT STATE.

2. **Initiation Invalid**: Return DONE if the Initiation condition is no longer satisfied by the CURRENT STATE.

3. **Otherwise**: Return CONTINUE if the skill should remain active.

**Response Requirements:** Output **EXACTLY ONE** line in the following format, with no extra text or explanation:

```
<status>DONE</status>  % or <status>CONTINUE</status>
```

---

### Prompt Template for Semantic Gradient Generation

You are a Skill Doctor. Your goal is to generate structured updates (semantic gradients) for a skill by diagnosing its execution history.

**Inputs:**

- **Skill Definition**: `{skill_info}` (Current Initiation, Policy, and Termination)

- **Execution Trace**: `{trajectory}` (The sequence of states and actions)

- **Result (Reward)**: `{reward}` (Indication of success or failure)

**Task Instructions:**

1. **Diagnosis**: Identify the ROOT CAUSE of the outcome.

2. **Prescription**: Map the identified cause to specific components for updates:
   - **Initiation**: Adjust if the skill was triggered in an inappropriate state.
   - **Policy**: Refine steps if the agent hallucinated, missed a transition, or chose wrong moves.
   - **Termination**: Update if the skill stopped prematurely or entered an infinite loop.

**Constraints:**

- Keep components as an empty string if no update is required.

- Set `is_related` to *False* only if the skill was completely irrelevant to the outcome.

- Ensure `semantic_gradient` provides concrete, actionable instructions (e.g., "Add a check for X").

**Response Requirements (JSON Format):**

```
{
    "diagnosis": "Brief explanation of the outcome...",
    "is_related": true/false,
    "semantic_gradient": {
        "initiation": "...",
        "policy": "...",
        "termination": "..."
    }
}
```

---

### Prompt Template for Skill Evolution (Optimization Step)

You are a Skill Evolver. Your goal is to apply a semantic gradient update to a skill based on aggregated feedback from execution traces.

**Inputs:**

- **Original Skill** ($\omega$): `{old_skill_definition}`

- **Semantic Gradients** ($g_i$): A collection of gradients detailing failures/successes in *Initiation*, *Policy*, and *Termination*.

**Task Instructions:**

1. **Batch-level Aggregation**: Identify systematic weaknesses across all gradients. Filter out noise and trajectory-specific details, focusing only on recurring patterns.

2. **Semantic Update**: Perform $\omega' = \omega \oplus \bar{g}$ to refine the skill:
   - **Initiation** ($I$): Refine the "IF" condition to ensure the skill only starts in valid states.
   - **Policy** ($\pi$): Update the 3–5 reasoning steps to bypass identified failure modes.
   - **Termination** ($\beta$): Update the "Stop IF" condition to strictly verify the outcome.

**Evolution Mode:**

- **REFINE**: Apply gradient ascent to improve the existing logic while keeping the core intent.

- **DISCOVER**: Synthesize a fundamentally NEW skill structure if the current one is irrelevant.

**Response Requirements (JSON Format):**

```
{
  "skill_name": "Concise_Name",
  "initiation": "IF... AND...",
  "policy": ["Step 1...", "Step 2...", "Step 3..."],
  "termination": "Stop IF..."
}
```

## Prompt Template for Trajectory Summarization (Ablation Study)

You are a trajectory compression assistant. Your goal is to provide a factual, neutral summary of an agent's execution trace without adding any diagnostic or prescriptive insights.
**Inputs:**

- **Trajectory**: {trajectory} (The raw sequence of states and actions)

- **Reward**: {reward} (The final outcome of the execution)

**Task Instructions:** Write a concise, factual summary of observable events based on the provided trajectory and reward.
**Operational Rules:**

1. **Facts Only**: Describe only what was observed in the execution.

2. **No Diagnosis**: Do not provide reasoning or explanations for failures or successes.

3. **No Advice**: Do not offer any suggestions, prescriptions, or improvements for the skill.

**Response Requirements (JSON Format):** Return ONLY one JSON object wrapped in triple backticks:

````
```json
{
  "summary": "..."
}
````

## Prompt Template for Skill Evolution without Semantic Gradient (Ablation Study)

You are an Evolution Operator. Your goal is to refine a skill based on a collection of neutral trajectory summaries, without the aid of causal diagnosis or semantic gradients.
**Inputs:**

- **Parent Skill ($\omega$)**: {old_skill_definition}

- **Trajectory Summaries**: A list of neutral, factual summaries of past execution traces and their corresponding rewards.

**Task Instructions:** Refine the parent skill based on the provided evidence. Focus on improving the success rate while preserving the core intent of the original skill.
**Operational Rules:**

1. **Evidence-based**: Use only observable facts from the summaries to justify changes.

2. **Strict Constraints**: Use only state-checkable terms in the conditions; avoid vague language like "successfully submitted" unless explicit feedback is present.

**Response Requirements (JSON Format):** Output **EXACTLY ONE** skill in strict JSON format. No explanations or extra text allowed.

```
{
  "skill_name": "Concise_Name",
  "initiation": "IF ... AND ... (fully checkable conditions)",
  "policy": [
    "S1: ...",
    "S2: ...",
    "S3: ..."
  ],
  "termination": "Stop IF ... (fully checkable conditions)"
}
```

# F. Discussion and Limitations

## F.1. From Explicit Skills to Implicit Procedural Memory

Our current instantiation of Skills differs from human procedural memory (Cohen & Squire, 1980; Squire, 2004) in that the latter is largely implicit and not directly observable, whereas our Skills are represented in an explicit, readable form. This explicitness is shared by existing agent skill systems, such as Claude Agent Skills (Anthropic, 2025), and facilitates learning, inspection, and system-level control. Importantly, explicit representation does not imply exposure to end users: in our framework, Skills can be concealed at the system level during execution and operate as internal procedural abstractions.

A natural direction for future work is to move beyond explicit representations toward genuinely implicit or directly executable Skills. One possible approach is to progressively compress frequently reused Skills into more compact forms, such as executable code modules, parameterized procedures, or latent control policies that can be invoked without natural-language mediation. Another direction is to decouple Skill execution from language generation entirely, allowing mature Skills to be executed directly while retaining explicit representations for learning, evaluation, and debugging. Through such mechanisms, explicit Skills may serve as an intermediate stage in skill acquisition, with long-term evolution yielding more implicit procedural representations that more closely resemble human procedural memory.

## F.2. API Compatibility of the PPO Gate

The PPO Gate in Section 4.2 requires only output-token log-probabilities. Open-weight models are fully compatible, and several closed-source API providers already expose this capability, including OpenAI (`logprobs` / `top_logprobs`), Gemini (`responseLogprobs` / `logprobs`), and Cohere (`logprobs`). The remaining compatibility constraint therefore stems from API exposure choices of specific providers, not from NP-PPO itself.

For providers that do not surface log-probabilities, a promising direction is to replace the importance ratio with purely prompt-based verification mechanisms, such as LLM-as-judge scoring or self-consistency filtering, to achieve full compatibility with completely opaque APIs.

