# OpenReview forum: "Skill-Pro: Learning Reusable Skills from Experience via Non-Parametric PPO for LLM Agents"
_ICML.cc/2026/Conference — ICML 2026 spotlight_

### Official Review · Reviewer_Rxd3 · 2026-03-13

**Soundness:** 3
**Presentation:** 3
**Significance:** 3
**Originality:** 3
**Overall Recommendation:** 5
**Confidence:** 3

**Summary:**

The paper formalizes procedural memory for llm agent as skills, defined as a temporally extended procedural unit specifying activation condition, execution procedure, and termination condition. And proposes Non-Parametric PPO to realize reliable skill pool evolution by defining semantic gradients, PPO-style trust region verification, and online score-based pruning. Experiments on ALFWorld and TextArena show that the proposed procedural memory is efficiently reused, the ablation study shows each design choice contributes to the final performance, and provides qualitative results.

**Compliance With Llm Reviewing Policy:**

Affirmed.

**Final Justification:**

I maintain my positive recommendation of the paper on the basis that the authors will incorporate all the additional discussions explicitly into the final paper.

**Key Questions For Authors:**

- What is the cost of skill evolution? How many tokens are needed for calculating the per-trajectory semantic gradients, llm-based batch-level aggregation, and llm-based gradient ascent step?
- How is the advantage estimated without training a value function? More details on this may help readers understand the method better.
- Every step in np-ppo are llm-based and potentially non-optimal. Can the theoretical correctness still be implied here?

**Limitations:**

- The procedural memory update might be costly.
- Computing the importance ratio requires access to the logits of LLM, making the method not applicable to black-box LLMs.
- Every step in np-ppo are llm-based and potentially non-optimal. Can the theoretical correctness still be implied here?

**Strengths And Weaknesses:**

Strengths:

- The formalization of Non-Parametric PPO is well-motivated and well-supported by the solid experiments.

- The paper is well-written.

- The analysis and ablation study are insightful.

- The qualitative results greatly helps understanding the benefits of the method.


Weaknesses:

- The cost of skill evolution is not discussed, and it seems to require lots of computation. How many tokens are needed for calculating the per-trajectory semantic gradients, llm-based batch-level aggregation, and llm-based gradient ascent step?
- Computing the importance ratio requires access to the logits of LLM, making the method not applicable to black-box LLMs.
- How is the advantage estimated without training a value function? More details on this may help readers understand the method better.
- Every step in np-ppo are llm-based and potentially non-optimal. Can the theoretical correctness still be implied here?
- For comparing different methods' effectiveness, not discussing the cost of memory update seems unfair.
- Some qualitative cases on how skills affect the low-level action output by LLMs may help the readers understand how the skills are helping the task overall.
- Minor:
  - Still ?? in line 670
  - l365: PorMem

---

> ### Author Rebuttal · Authors · 2026-03-31
>
> **Dear Reviewer Rxd3**,
>
> Thank you for your thoughtful review and for recognizing our work as “well-motivated,” “well-written,” with “solid experiments” and “insightful” analysis.
>
> During the rebuttal period, we addressed your concerns through additional experiments, targeted clarifications, and examples:
>
> - **Added the cost of skill evolution.**
> - **Clarified that some providers (e.g. OpenAI) expose log-probability APIs.**
> - **Pointed to the advantage estimation in the paper.**
> - **Clarified the positioning of NP-PPO**.
> - **Added a qualitative case showing how skills affect low-level LLM actions.**
>
> Below, we provide detailed responses to each of your questions, with the utmost sincerity, to fully address every concern.
>
> ---
>
> > **W1 & W5 & Q1: "The cost of skill evolution is not discussed..." "What is the cost of skill evolution? "**
>
> **Answer**: Since learned skills are meant for repeated reuse across future tasks and agents, Table 1 focuses on deployment-time efficiency. To address this concern, **we added the cost of skill evolution** in ProcMEM, as shown below. One evolution round includes per-trajectory semantic-gradient computation and one LLM call for batch aggregation plus semantic update. These costs are paid during learning, while **the learned skills can be reused many times afterward**. We will add this in the revised paper.
>
> ||Input |Output |Total Tokens|
> |-|-:|-:|-:|
> |Per-traj semantic gradients|1156.0±45.5|106.7±10.4|1262.7±46.7|
> |Batch aggregation+semantic update|670.0±51.9|159.6±18.7|829.6±55.7|
> |Tokens/round(6 traj)|7606.0|799.8|8405.8|
>
> ---
>
> > **W2: "Computing the importance ratio requires access to the logits of LLM, making the method not applicable to black-box LLMs."**
>
> **Answer**: The importance ratio in Eq.6 requires output-token log-probabilities, but not access to model weights. This is already **supported** by several mainstream providers, including **OpenAI** (`logprobs` / `top_logprobs`), **Gemini** (`responseLogprobs` / `logprobs`), and **Cohere** (`logprobs`). Therefore, the limitation for some black-box LLMs is due to API exposure, **not** the NP-PPO itself. We will clarify this scope in the revision.
>
> ---
>
> > **W3 & Q2: "How is the advantage estimated without training a value function?..."**
>
> **Answer: As stated in the paper(p.5, line 230)**, we estimate the advantage as return-to-go with a running baseline, $\hat A_t = G_t - \bar R$. This estimate is used in PPO Gate to score candidate skills on historical trajectories, without training a separate value function.
>
> ---
>
> > **W4 & Q3: "Every step in np-ppo are llm-based and potentially non-optimal. Can the theoretical correctness still be implied here?"**
>
> **Answer**: We do **not** claim NP-PPO inherits the theoretical guarantees of standard PPO. **As stated in the paper(line 266-268)**, semantic-gradient updates are LLM-generated and may "introduce hallucinated or behaviorally unstable Skills".
>
> **Precisely for this reason, we introduce PPO Gate** to filter candidate skills before adding them to the pool. Thus, NP-PPO is a PPO-inspired non-parametric framework, where PPO Gate acts as a practical trust-region safeguard. Empirically, removing PPO Gate degrades performance and stability (Table 3, Fig. 3).
>
> ---
>
> > **W6: "Some qualitative cases on how skills affect the low-level action output by LLMs..."**
>
> **Answer**: Thank you for helpful suggestions. We agree that qualitative cases improve interpretability. **We added a representative case** from ALFWorld:
>
> **State**
>
> ```text
> You are in the middle of a room. Looking quickly around you, you see a microwave 1, a cabinet 3,...(remaining details omitted)
>
> Your task is to: clean some tomato and put it on countertop.
>
> > go to fridge 1
> > You arrive at loc 7. The fridge 1 is closed.
> ```
> **Activated skill: `SelfConsistencyCheck`**
>  - Initiation: When the action must strictly satisfy known rules or historical constraints.
>  - Procedure:
>    1. Propose a candidate action.
>    2. Check whether this action violates any known rules or past feedback.
>    3. If a violation is found, revise the action to remove the violation.
>    4. Repeat the check until no violation remains.
>    5. Output the final consistent action.
>  - Termination: An action that passes all self-consistency checks is produced.
>
> |Setting|Output action|
> |-|-|
> |Without skill| `take tomato 1 from fridge 1` |
> |With skill | `open fridge 1` |
>
> The feedback imposes a constraint (fridge is closed). `SelfConsistencyCheck` rejects invalid candidate actions and revises them to the valid prerequisite (open fridge 1), showing how skills steer low-level LLM actions. We will add this in the revised paper.
>
> ---
>
> > **W7: "Minor..."**
>
> **Answer**：Thank you for pointing this out. We will revise these issues in the paper.
>
> ---
> **We hope these clarifications and new results will encourage you to reconsider your score. If you have any further questions, we would be very happy to discuss them.**

---

> > ### Author Rebuttal · Reviewer_Rxd3 · 2026-04-02
> >
> > Thanks for providing the additional clarifications, cost analysis for skill evolution, and a representative case. However, part of my concerns remains:
> >
> > - W5: The performance gain and efficiency tradeoff should be discussed explicitly in the paper; still no comparison on the cost of different baselines.
> > - W2: I agree some proprietary models may expose the logprob api, but still, this is a method limitation and should be explicitly discussed in the paper.
> > - W6: This provided case actually raises some additional doubts on the efficiency of the method.
> >   - More advanced models may not make this mistake even without skills, which makes the revocation of this skill wasteful somehow.
> >   - Even with a failed attempt with `take tomato 1 from fridge 1` and feedback, and realize need to open first might actually be cheaper than running the SelfConsistencyCheck skill?
> >
> > Overall, I do not think these limitations outweigh the merits of this paper, so I maintain my positive review, but these should be discussed explicitly in the paper rather than being hidden.

---

> > > ### Author Response · Authors · 2026-04-03
> > >
> > > Dear Reviewer Rxd3,
> > >
> > > **Thank you for maintaining your positive overall recommendation of 5 (Accept). We really appreciate your helpful follow-up and agree that these mentioned points should be discussed more explicitly in our paper.**
> > >
> > > To fully address this, we provide additional experimental results and clarifications below.
> > >
> > > > **"W5: The performance gain and efficiency tradeoff should be discussed explicitly in the paper; still no comparison on the cost of different baselines."**
> > >
> > > **Answer:** We agree that the performance–efficiency trade-off should be discussed more explicitly.
> > >
> > > - In the submission, **Table 1 reports reuse performance together with deployment-time efficiency, including storage cost and execution cost.** In the rebuttal, we **added ProcMEM’s memory-evolution cost (i.e., the skill-evolution cost)**.
> > > - ProcMEM achieves substantially better reuse and lower deployment cost, but at the price of an upfront offline evolution cost. This trade-off is most meaningful when the learned skills are reused repeatedly across future tasks or agents.
> > > - To address this concern more directly, we **further added the average memory-evolution cost** for a strong baseline with an explicit update mechanism, G-Memory, using the same token accounting protocol, as shown below.
> > > - Under this comparison, **ProcMEM not only improves reuse and deployment-time efficiency, but also has substantially lower memory-evolution cost than G-Memory.**
> > >
> > > We will clarify this trade-off explicitly in the revised paper and add the costs of the other baselines.
> > >
> > > | Method   | Input Tokens | Output Tokens | Total Evolution Tokens |
> > > | -- | --- | - | --- |
> > > | G-Memory | 21676.3±11247.1  | 1249.4±766.2| 22925.7± 12013.3  |
> > > | ProcMEM  | 7606.0 ± 275.9   | 799.8 ±75.9  | 8405.8 ±307.4 |
> > >
> > > ---
> > >
> > >
> > >
> > > > **"W2: I agree some proprietary models may expose the logprob api, but still, this is a method limitation and should be explicitly discussed in the paper."**
> > >
> > > **Answer:** We agree that requiring logprob access is a limitation of the current NP-PPO instantiation, especially for APIs that do not expose it. At the same time, this requirement is still weaker than that of standard PPO, since our method relies only on output-token log-probabilities rather than model weights or gradients. We will make this explicit in the revised paper.
> > >
> > > ---
> > >
> > >
> > >
> > > > "**W6: This provided case actually raises some additional doubts on the efficiency of the method.**"
> > >
> > > **Answer:** We agree with this concern and will state it explicitly in the revised paper:
> > >
> > > - The qualitative case is intended to show how a learned skill changes the low-level action, not to demonstrate efficiency.
> > > - For stronger base models or simpler states/tasks, the marginal benefit of such a skill may indeed be smaller.
> > > - The value of a skill lies not in making every single step cheaper, but in **being reusable across recurring situations** where it can improve overall task performance.
> > >
> > > ***
> > >
> > > Again, thank you for your valuable suggestions which have undoubtedly contributed to improving the quality of our paper.
> > >
> > > Many thanks,
> > >
> > > The authors of #12341

---

### Official Review · Reviewer_9f2D · 2026-03-13

**Soundness:** 3
**Presentation:** 3
**Significance:** 3
**Originality:** 3
**Overall Recommendation:** 4
**Confidence:** 3

**Summary:**

This paper proposes ProcMEM, a framework that enables LLM agents to learn reusable procedural memory from interaction experience without updating model parameters. The core idea is to formalize "Skills" as structured procedural units with activation, execution, and termination conditions, embedded in a Skill-MDP. A Non-Parametric PPO mechanism uses semantic gradients from trajectories to generate refined skill candidates, and a PPO Gate filters them via trust-region verification. Experiments on ALFWorld and TextArena Mastermind show improved reuse rates, cross-task and cross-agent generalization, and significant memory compression over episodic baselines.

**Compliance With Llm Reviewing Policy:**

Affirmed.

**Final Justification:**

The rebuttal has addressed my concerns so that I decide to increase the score.

**Key Questions For Authors:**

1. How exactly are the importance ratios in Eq. 6 computed in practice? Do you require white-box access to the LLM for log-probabilities, or is there an approximation? This is critical for understanding the actual applicability of the PPO Gate.

2. The standard deviations in Table 2 for ProcMEM are noticeably larger than those of baselines. What drives this instability, and does it diminish with more training iterations or a larger skill pool?

3. How sensitive is performance to the initial skill seeds? The paper mentions "fixed skill seeds" for the w/o NP-PPO ablation but does not discuss how the seeds are chosen or whether poor initialization could cause the framework to converge to a suboptimal skill pool.

4. Have you tested on tasks with substantially different structure from the two benchmarks used, such as open-ended web navigation or tool-use environments? The current evaluation scope makes it hard to assess how well Skills generalize beyond well-defined game settings.

If the author can address my concern, I would be willing to increase my score.

**Limitations:**

yes

**Strengths And Weaknesses:**

Strengths:
- The Skill-MDP formalization is clean and provides a principled way to integrate procedural memory into LLM agent decision loops.
- The analogy to PPO is creative: semantic gradients for candidate generation and a clipped surrogate gate for verification is a neat non-parametric counterpart.
- Memory compression is striking, with only 816 tokens stored versus hundreds of thousands for baselines, while maintaining competitive or better performance.
- The evolutionary lineage visualization and skill distribution analysis across agents and difficulties provide useful transparency into what the system actually learns.

Weaknesses:
- Evaluation is limited to two benchmarks, one of which is Mastermind, a relatively narrow game. Claims about general long-term autonomy need broader task diversity to be convincing.
- The high variance in ProcMEM results across Table 2 is concerning and insufficiently discussed. For instance, ALFWorld Train shows 0.900 plus or minus 0.300, which means the lower end barely outperforms some baselines.
- The importance ratio computation in Eq. 6 assumes access to token-level log-probabilities under both old and new skill conditioning, but the paper does not clarify how this is obtained for black-box or API-only LLMs, limiting practical applicability.

---

> ### Author Rebuttal · Authors · 2026-03-31
>
> **Dear Reviewer 9f2D**,
>
> Thank you for your thoughtful review and for recognizing our work as a “clean” formulation, with a “creative” method and “competitive” performance.
>
> During the rebuttal period, we addressed your concerns through additional experiments, targeted clarifications, and examples:
>
> - **Added experiments on a more complex benchmark, BFCL.**
> - **Added new statistics over multiple runs.**
> - **Clarified that some providers (e.g. OpenAI) expose log-probability APIs.**
> - **Added experiments to analyze sensitivity to initial skill seeds.**
>
> Below, we provide detailed responses to each of your questions, with the utmost sincerity, to fully address every concern.
>
> ---
>
>
>
> > **W1 & Q4: "Evaluation is limited to two benchmarks,..."**
>
> **Answer:** Thank you for this important comment. We would like to clarify that **the paper already includes ALFWorld (Table 2, Fig. 6), which goes beyond narrow game**: the agent must choose from a large, state-dependent admissible action set, with dozens of admissible actions (e.g., 35–36 in consecutive states) and a dynamically changing action space.
>
> **We added experiments on BFCL v4** (Berkeley Function Calling Leaderboard), a benchmark with more complex, multi-turn, and larger function-space decision scenarios. As shown below, ProcMEM outperforms the compared baselines on this more diverse benchmark, with low deployment cost. We will include broader baseline comparisons in the revised paper.
>
> |Baselines|Accuracy ↑|
> |-|-:|
> |State|0.233±0.025|
> |ReAct|0.383±0.023|
> |CoT|0.367±0.023|
> |ProcMEM(Ours)|0.433±0.050|
>
> ProcMEM’s skill deployment cost:
> | Total Stored Tokens | Avg. Tokens /Unit | Retrieval Ratio | Δ Prompt Tokens / Step |
> |-:|-:|-:|-:|
> |489.500±14.500|69.930±2.070|0.379±0.007|178.060±1.850|
>
>
>
> ---
>
> > **W2 & Q2: "The high variance in ProcMEM results across Table 2 is concerning and insufficiently discussed..."**
>
> **Answer**: We agree that the large variance for ProcMEM on ALFWorld-Train in Table 2 was not explained clearly enough. The originally reported `±0.300` reflected episode-level dispersion of a binary success metric, which is naturally high and mainly captures task-difficulty variation rather than method stability. In fact, **statistics over independent runs are more appropriate**, since they reflect method-level randomness and reproducibility.
>
> We therefore **added new statistics** over independent runs. Over 10 repeated runs with 5 episodes each (50 episodes total), ProcMEM achieves 0.900 ± 0.105 (SE = 0.033, 95% CI = [0.825, 0.975]). Importantly, the mean is unchanged, indicating that the issue lies in the uncertainty summary, not the underlying performance.
>
> We also **added the table** below to show how the estimate changes with the number of runs. The mean remains 0.900 while the uncertainty steadily decreases, indicating that the original `±0.300` mainly came from episode-level noise, rather than genuine instability of ProcMEM. We will clarify this in the revised paper.
>
>
> |Repeated runs|Total episodes|Mean|Std|SE|95%CI|
> |-:|-:|-:|-:|-:|-:|
> |4|20|0.900|0.115|0.058|[0.716,1.000]|
> |6|30|0.900|0.110|0.045|[0.785,1.000]|
> |8|40|0.900|0.107|0.038|[0.811,0.989]|
> |10|50|0.900|0.105|0.033|[0.825,0.975]|
>
> ---
>
> > **W3 & Q1: "The importance ratio computation in Eq. 6 assumes access to token-level log-probabilities under both old and new skill conditioning,..."**
>
> **Answer**: The importance ratio in Eq.6 requires output-token log-probabilities, but not access to model weights. This is already **supported** by several mainstream providers, including **OpenAI** (`logprobs` / `top_logprobs`), **Gemini** (`responseLogprobs` / `logprobs`), and **Cohere** (`logprobs`). Therefore, the limitation for some black-box LLMs is due to API exposure, **not** the NP-PPO itself. We will clarify this scope in the revision.
>
> ---
>
> > **Q3: "How sensitive is performance to the initial skill seeds? ..."**
>
> **Answer**: We agree that initialization sensitivity is important. We therefore **added experiments with a weaker seed initialization**, constructed by removing the execution procedures from our default seeds. As shown below, NP-PPO still improves both performance and reuse under weaker seeds, showing that it can effectively evolve skills beyond the initial seed pool.
>
> At the same time, **stronger initial seeds lead to better final results**, which is consistent with intuition, since the user-defined seed pool serves as the starting point for subsequent skill evolution.
>
> |Init.|Reuse Rate ↑|Performance ↑|
> |-|-:|-:|
> |Default seeds+NP-PPO|0.925±0.061|0.606±0.234|
> |Default seeds w/o NP-PPO|0.563±0.176|0.482±0.197|
> |Weaker seeds+NP-PPO|0.488±0.170|0.481±0.209|
> |Weaker seeds w/o NP-PPO|0.163±0.057|0.438±0.151|
>
> ---
>
> **We hope these clarifications and new results will encourage you to reconsider your score. If you have any further questions, we would be very happy to discuss them.**

---

> > ### Author Rebuttal · Reviewer_9f2D · 2026-04-02
> >
> > Thanks for the rebuttal and I have increased my score

---

> > > ### Author Response · Authors · 2026-04-04
> > >
> > > We really appreciate that your concerns have been adequately addressed and we are truly grateful for your dedicated time and constructive comments!

---

### Official Review · Reviewer_Bxdr · 2026-03-13

**Soundness:** 3
**Presentation:** 3
**Significance:** 3
**Originality:** 3
**Overall Recommendation:** 4
**Confidence:** 3

**Summary:**

This paper introduces ProcMEM, a reusable skill pool management framework to improve sequential decision-making of LLM agents. To evolve skill pool without parameter updates, ProcMEM introduces non-parametric PPO. First, based on trajectory, LLM is asked to propose structured refinements to each skill, then a PPO-style trust-region gate filters individual candidates to be added to the skill pool to prevent excessive behavioral changes. The score-based pruning then maintains a compact skill pool based on skill score and semantic redundancy between skills. Experiments show that ProcMEM outperforms baselines while maintaining huge memory compression.

**Compliance With Llm Reviewing Policy:**

Affirmed.

**Final Justification:**

The rebuttal has addressed my concerns, particularly regarding scalability and stability. I have increased my rating to positive.

**Key Questions For Authors:**

- How is the initial skill pool initialized?
- In Table 4, the PPO Gate Pass Rate looks strange. The PPO gate may be filtering all or nothing per batch, rather than providing consistent selective pressure.

**Limitations:**

The framework is only validated in simple environments with narrow action spaces. The paper lacks analysis or insights on how the framework might behave in more complex settings.

**Strengths And Weaknesses:**

### Strengths
- The problem formulation is refreshing and well-motivated.
- The paper introduces a novel method, Non-Parametric PPO with clear description of each process to update the skill pool.
- Automatic skill management is timely given the emergence of skill systems for LLM agents.

### Weaknesses
- Unlike parametric PPO, the LLM-based semantic update may produce hugely different candidates from the original skill. It remains unclear whether PPO gating is sufficient to prevent destabilizing shifts.
- Experiments are limited to environments with narrow action spaces. It is unclear whether the framework scales to more complex environments where the search space at the language level is far larger.

---

> ### Author Rebuttal · Authors · 2026-03-31
>
> **Dear Reviewer Bxdr**,
>
> Thank you for your thoughtful review and for recognizing our work as “refreshing and well-motivated,” with a "novel method", "clear description", and timely relevance to skill systems for LLM agents.
>
> During the rebuttal period, we addressed your concerns through additional experiments, targeted clarifications, and examples:
>
> - **Clarified that PPO Gate is designed to filter hugely different, hallucinated, or behaviorally unstable skill candidates.**
> - **Clarified that the paper already includes ALFWorld, which goes beyond narrow fixed action spaces.**
> - **Added experiments on a more complex benchmark, BFCL.**
> - **Added representative examples of the initial skill pool.**
> - **Clarified that PPO Gate Pass Rate is defined over the full skill-evolution process.**
>
> Below, we provide detailed responses to each of your questions, with the utmost sincerity, to fully address every concern.
>
> ---
>
> > **W1: "Unlike parametric PPO, the LLM-based semantic update may produce hugely different candidates..."**
>
> **Answer**:
>
> - We agree that semantic updates may generate candidates that differ substantially from the original skill. In fact, **the paper already states this** explicitly (Sec.4.2 Lines 266–268), noting that semantic-gradient updates may introduce “hallucinated or behaviorally unstable Skills.”
> - **This is precisely why we introduce the PPO Gate**. If a candidate induces an overly large behavioral shift, its action-likelihood ratios on historical trajectories will deviate markedly from 1, and the clipped PPO-style objective will lower its verification score, reducing its chance of being accepted. Thus, PPO Gate serves as a practical trust-region-style safeguard.
> - This is also supported by the ablations: **Table 3 and Fig. 3 already show** that removing PPO Gate reduces both performance and stability.
>
>
>
> > **W2: "Experiments are limited to environments with narrow action spaces..."**
>
> **Answer:** Thank you for this important comment. We would like to clarify that **the paper already includes ALFWorld (Table 2, Fig. 6), which goes beyond narrow game**: the agent must choose from a large, state-dependent admissible action set, with dozens of admissible actions (e.g., 35–36 in consecutive states) and a dynamically changing action space.
>
> **We added experiments on BFCL v4** (Berkeley Function Calling Leaderboard), a benchmark with more complex, multi-turn, and larger function-space decision scenarios. As shown below, ProcMEM outperforms the compared baselines on this more diverse benchmark, with low deployment cost. We will include broader baseline comparisons in the revised paper.
>
> |Baselines|Accuracy↑|
> |-|-:|
> |State|0.233±0.025|
> |ReAct|0.383±0.023|
> |CoT|0.367±0.023|
> |ProcMEM(Ours)|0.433±0.050|
>
> ProcMEM’s skill deployment cost:
> |Total Stored Tokens|Avg.Tokens/Unit|Retrieval Ratio|ΔPrompt Tokens/Step|
> |-:|-:|-:|-:|
> |489.5±14.5|69.93±2.07|0.379±0.007|178.06±1.85|
>
>
>
>
> > **Q1: "How is the initial skill pool initialized?"**
>
> **Answer**: The initial skill pool is manually initialized with a small set of **general** procedural skills, rather than benchmark-specific ones. It includes StructuredCoT, ReActDecision, HypothesisElimination, SelfConsistencyCheck, ExploreExploitArbitration, and StrategicPlanning.
>
> Due to space limits, we **showed** two representative examples below and will add the full initialization details in the revised paper.
>
> **StructuredCoT:**
>
> ```text
> - Activation: When a decision must be made based on multiple constraints or past feedback.
> - Execution:
>   1. Restate the immediate goal of the task in one sentence.
>   2. List all hard constraints implied by the current state and feedback history.
>   3. Summarize the key information from previous actions and feedback that affects the decision.
>   4. Compare the main candidate actions step by step under these constraints.
>   5. Select the single action that best satisfies the constraints and goal.
> - Termination: A single concrete action is selected and output.
> ```
> **ReActDecision:**
>
> ```text
> - Activation: When the environment provides feedback after each action and past feedback should influence the next decision.
> - Execution:
>   1. Interpret the most recent feedback and explain what it implies about the environment.
>   2. Update your belief about which actions or outcomes are more or less likely.
>   3. Choose the next action that best exploits or tests this updated belief.
>   4. Output only the chosen action.
> - Termination: The next action is selected by the updated belief.
> ```
>
> > **Q2: "In Table 4, the PPO Gate Pass Rate looks strange..."**
>
> **Answer**: Thank you for pointing this out. The PPO Gate Pass Rate is the **average pass rate over the full skill-evolution process**, not an all-or-nothing decision per batch. We will clarify this definition in the revised paper.
>
> ---
>
> **We hope these clarifications and new results will encourage you to reconsider your score. If you have any further questions, we would be very happy to discuss them.**

---

> > ### Author Rebuttal · Reviewer_Bxdr · 2026-04-03
> >
> > Most of my concerns have been resolved. Thank you for the clarification and the additional experimental result. I will increase my rating.

---

> > > ### Author Response · Authors · 2026-04-03
> > >
> > > Dear Reviewer Bxdr,
> > >
> > > We really appreciate your recognition of our clarification and the additional experimental result, and your concerns have been adequately addressed. Thank you for upgrading your rating and your valuable suggestions!
> > >
> > > Many thanks,
> > >
> > > The authors of #12341

---

### Official Review · Reviewer_J8Sm · 2026-03-14

**Soundness:** 3
**Presentation:** 3
**Significance:** 3
**Originality:** 3
**Overall Recommendation:** 4
**Confidence:** 4

**Summary:**

This paper presents ProcMEM, turning  agent’s past experience into reusable procedural skills.  It introduces Non-Parametric PPO, which leverages semantic gradient for candidate skill generation, enabling agents to autonomously learn procedural memory from interaction. The experiments are mainly on TextArena Mastermind variants and ALFWorld, showing good task performance and much more compact memory than several episodic-memory style  baselines.

**Compliance With Llm Reviewing Policy:**

Affirmed.

**Final Justification:**

Rebuttal answered many of my concerns, and considering other people's ratings, I'm raising the score to weak accept.

**Key Questions For Authors:**

1. Why not compare the PPO-style gate against simpler candidate-filtering baselines? A direct comparison to simpler validation strategies would make it much easier to assess whether the PPO-style design is necessary, or whether most of the gain comes from candidate filtering in general.
2. Could the reuse-rate metric be misleading? Since reuse is measured as the fraction of skills invoked at least once, it seems naturally easier to score well with a small skill pool. It would help to also report things like invocation frequency distribution, effective coverage, or marginal utility per skill under matched memory budgets.

**Limitations:**

Overall, I buy the motivation, and I think the paper has a good underlying idea. The empirical results are promising. But for the paper to be fully convincing, it needs cleaner positioning against the closest procedural memory work, a more careful description of what the PPO-style gate actually does, and a more honest end-to-end accounting of efficiency.

**Strengths And Weaknesses:**

##### Strengths

- I think the paper is tackling a real problem. Long-horizon agents cannot just keep appending interaction history forever.The idea of representing memory as explicit reusable skills is clean and intuitive.
- The concept of Non-Paramtic PPO is attractive. The use of semantic gradients for candidate generation and a gating mechanism for robust skill verification is reasonable.
- Experiments show that ProcMEM achieves superior reuse rates and significant performance gains with extreme memory compression.

- The ablations are meaningful, showing that removing the gate or the online scoring mechanism clearly hurts performance, which suggests the system is more than just a prompt trick.



##### Weaknesses

- The paper argues that temporally extended skills reduce retrieval frequency, but Table 1 seems to show ProcMEM retrieving more often than some baselines (AWM, G-Memory).

- The efficiency evaluation is mostly based onmetrics such as stored tokens or prompt-token overhead. That is useful, but it is not the same as measuring real end-to-end cost. There is no wall-clock latency or full accounting of the cost of maintaining and evolving the skill pool.

- The PPO framing feels a bit stronger than what is actually happening. As far as I understand, this is not PPO training in the usual sense, but a clipped-ratio filtering mechanism applied to candidate skill rewrites on old trajectories. That may still be a good design, but the paper should present it more carefully.

- The positioning against closely related work is still not sharp enough. This paper sits very close to recent lines on procedural, workflow, or structured memory. The novelty is there, but the boundary with nearby methods needs to be explained more clearly and ideally backed by tighter comparisons.

- Typos:  some citations in the appendix are  question marks (line 670).

---

> ### Author Rebuttal · Authors · 2026-03-31
>
> **Dear Reviewer J8Sm**,
>
> Thank you for your thoughtful review and for recognizing our work as “tackling a real problem,” with an “good underlying idea”, “promising” results, and “meaningful” ablations.
>
> During rebuttal, we addressed your concerns through targeted clarifications and additional experiments:
>
> - **Clarified that ProcMEM reduces unnecessary retrieval and enables finer-grained reuse.**
> - **Added the token cost of skill evolution.**
> - **Clarified NP-PPO’s PPO-inspired positioning (Sec.4, Lines195–200).**
> - **Added a comparison table against workflow and structured memory.**
> - **Added results for a simpler filtering baseline.**
> - **Pointed to Fig. 5 for the Skill Selection Distribution.**
> - **Clarified that the compact skill pool results from learning reusable skills, not the cause of high reuse.**
>
> Below, we provide detailed responses to address each concern.
>
> ---
>
> > **W1: "The paper argues that temporally extended skills reduce retrieval frequency,..."**
>
> **Answer:** We clarify three points:
>
> - Lower retrieval frequency is **not** our main claim; our goal is to learn reusable procedural skills, not to minimize retrieval count.
> - ProcMEM retrieves a skill only when its Activation Condition is satisfied by the current state, thereby reducing **unnecessary retrieval compared with step-level retrieval**.
> - AWM and G-Memory retrieve only once per episode/task, so ProcMEM may retrieve more often than these methods, but it enables finer-grained reuse.
>
> ---
>
> > **W2: "The efficiency evaluation is mostly based on metrics..."**
>
> **Answer:**
>
> - Since the value of skills lies in their reuse across future tasks and agents, Table 1 reports deployment-time efficiency.
> - The costs of learning and maintaining the skill pool are paid during training, while learned skills are reused many times afterward.
> - **We added the token cost of skill evolution** to provide a fuller efficiency picture.
>
> ||Input|Output|Total Tokens|
> |-|-:|-:|-:|
> |Per-traj semantic gradients|1156.0±45.5|106.7±10.4|1262.7±46.7|
> |Batch aggregation+semantic update|670.0±51.9|159.6±18.7|829.6±55.7|
>
> ---
>
> > **W3: "The PPO framing feels a bit stronger than what is actually happening..."**
>
> **Answer: As stated in Sec.4(line 195–200),** NP-PPO is a PPO-inspired non-parametric method for skill evolution, not standard PPO training: it does not update policy parameters, but generates candidate skills via semantic gradients and verifies them with PPO-style clipped-ratio verification on historical trajectories under a frozen LLM policy.
>
> ---
>
> > **W4: "The positioning against closely related work is still not sharp enough..."**
>
> **Answer:** ProcMEM is procedural memory with skills as the memory unit. **We added the following comparison table** to clarify its distinction from related works.
>
> |Aspect|Workflow memory|Structured memory|ProcMEM|
> |-|-|-|-|
> |Memory unit|Workflows from successful trajectories|Compressed episodic traces, e.g., notes/reflections/summaries|Skills as procedural memory|
> |Trigger|Current task|Current query/context via retrieval|**Recurring situation** via Activation Condition|
> |Role|Task-level execution|Store/retrieve past experience for context|Reusable procedures across tasks|
> |Update|Workflow induction+memory integration|Accumulation/compression/reflection/graph update|PPO-inspired non-parametric skill evolution|
> |Generalization|task/subtask-level|Usually experience/context-level|Situation-level, task-agnostic procedural reuse|
>
> ---
>
> > **W5: "Typos..."**
>
> **Answer:** We will fix them in revision.
>
> ---
>
> > **Q1: "Why not compare the PPO-style gate against simpler candidate-filtering baselines?..."**
>
> **Answer: We added results** under a simpler LLM-judge filter, which directly asks the LLM whether a candidate skill should enter the pool.
>
> Results show that PPO Gate achieves higher reuse and performance than both the LLM-judge filter and the no-filter variant, **indicating** that most of the gain comes from PPO Gate rather than filtering alone.
>
> |Filter|Reuse rate|Performance|Online score|Filter Pass Rate|
> |-|-:|-:|-:|-:|
> |PPO Gate|0.925±0.061|0.606±0.234|0.0406±0.0022|59.49%±49.09%|
> |LLM-judge|0.240±0.102|0.515±0.204|0.0536±0.0307|37.50%±33.07%|
> |No-filter|0.222±0.083|0.453±0.167|0.0011±0.0033|100.00%±0.00%|
>
> ---
>
> > **Q2: "Could the reuse-rate metric be misleading?..."**
>
> **Answer:**
>
> - **Fig. 5 already shows the Skill Selection Distribution**, i.e., invocation frequency distribution. Thus, reuse rate is not misleading: it reflects both frequent and low-frequency but necessary skills.
> - ProcMEM is designed to learn high-quality, reusable skills, and therefore naturally yields a compact skill pool; **the compact pool is a result**, not the cause, of high reuse.
> - Table 1 already compares memory budgets. ProcMEM uses only hundreds of tokens while baselines require tens or hundreds of thousands.
>
> **We hope these clarifications and new results encourage you to reconsider your score, and we would be happy to address any further questions.**

---

### Decision · Program_Chairs · 2026-04-30

**Decision:**

Accept (spotlight)

**Comment:**

This paper proposes ProcMEM, a framework enabling LLM agents to autonomously learn reusable procedural memory from interaction experience without parameter updates. The core contributions are: (1) a Skill-MDP formalization that structures skills with activation, execution, and termination conditions, and (2) Non-Parametric PPO (NP-PPO), which uses semantic gradients for candidate skill generation and a PPO-style trust-region gate (PPO Gate) for skill verification and pool maintenance.

The problem motivation is well-founded: long-horizon agents cannot rely solely on growing episodic memory. The formalization of procedural memory as structured, reusable skills is meaningful. The NP-PPO mechanism is novel, drawing a principled analogy to PPO while operating entirely in the non-parametric, language-based regime. Memory compression results are encouraging. ProcMEM stores only hundreds of tokens versus tens or hundreds of thousands for episodic baselines, with competitive or superior task performance.

Concerns regarding the limited scope of evaluation were substantially addressed through the inclusion of BFCL v4 results, which demonstrate generalization to multi-turn settings and larger function spaces. The performance–efficiency tradeoff and comparisons with relevant baselines should be explicitly incorporated into the final manuscript. The clarification regarding the log-probability access requirement should also be included in the revision. The authors are encouraged to clarify the conditions under which ProcMEM's procedural memory yields benefits beyond what a capable base model can achieve on its own.

Overall, the paper makes a timely contribution to the problem of experience reuse in LLM agents. This work will be of value to the community.